# Missense mutation of *Fmr1* results in impaired AMPAR-mediated plasticity and socio-cognitive deficits in mice

Marta Prieto[1,6], Alessandra Folci[1,6], Gwénola Poupon[1], Sara Schiavi[2], Valeria Buzzelli[2], Marie Pronot[1], Urielle François [3], Paula Pousinha[1], Norma Lattuada[4], Sophie Abelanet[1], Sara Castagnola[1], Magda Chafai[1], Anouar Khayachi[1], Carole Gwizdek[1], Frédéric Brau[1], Emmanuel Deval[1], Maura Francolini[4], Barbara Bardoni [5], Yann Humeau [3], Viviana Trezza[2] & Stéphane Martin [5✉]

Fragile X syndrome (FXS) is the most frequent form of inherited intellectual disability and the best-described monogenic cause of autism. CGG-repeat expansion in the *FMR1* gene leads to *FMR1* silencing, loss-of-expression of the Fragile X Mental Retardation Protein (FMRP), and is a common cause of FXS. Missense mutations in the *FMR1* gene were also identified in FXS patients, including the recurrent FMRP-R138Q mutation. To investigate the mechanisms underlying FXS caused by this mutation, we generated a knock-in mouse model (*Fmr1^{R138Q}*) expressing the FMRP-R138Q protein. We demonstrate that, in the hippocampus of the *Fmr1^{R138Q}* mice, neurons show an increased spine density associated with synaptic ultra-structural defects and increased AMPA receptor-surface expression. Combining biochemical assays, high-resolution imaging, electrophysiological recordings, and behavioural testing, we also show that the R138Q mutation results in impaired hippocampal long-term potentiation and socio-cognitive deficits in mice. These findings reveal the functional impact of the FMRP-R138Q mutation in a mouse model of FXS.

---

[1] Université Côte d'Azur, CNRS, IPMC, Valbonne, France. [2] RomaTre University, Dept. Science, Rome, Italy. [3] University of Bordeaux, CNRS, IINS, Bordeaux, France. [4] Università degli Studi di Milano, Dept. of Medical Biotechnology and Translational Medicine, Milan, Italy. [5] Université Côte d'Azur, Inserm, CNRS, IPMC, Valbonne, France. [6] These authors contributed equally: Marta Prieto, Alessandra Folci. ✉email: martin@ipmc.cnrs.fr

The formation of functional synapses in the developing brain is fundamental to establishing efficient neuronal communication and plasticity, which underlie cognitive processes. In the past years, synaptic dysfunction has clearly emerged as a critical factor in the etiology of neurodevelopmental disorders including Autism Spectrum disorder (ASD) and Intellectual Disability (ID). X-linked ID (XLID) accounts for 5–10% of ID patients and is caused by mutations in genes located on the X chromosome. The Fragile X Syndrome (FXS) is the most frequent form of inherited XLID and the first monogenic cause of ASD with a prevalence of 1:4000 males and 1:7000 females[1]. The majority of FXS patients exhibit mild-to-severe ID associated with significant learning and memory impairments, Attention Deficit Hyperactivity Disorder (ADHD), and autistic-like features[2–4]. To date, no effective therapeutic strategies are available.

FXS generally results from a massive expansion of the trinucleotide CGG (> 200 repeats) in the 5′-UTR region of the *FMR1* gene leading to its transcriptional silencing and consequently, the lack of expression of the encoded Fragile X Mental Retardation Protein (FMRP)[2,5]. FMRP is an RNA-binding protein that binds a large subset of mRNAs in the mammalian brain and is a key component of RNA granules. These granules transport translationally-repressed mRNAs essential for the synaptic function along axons and dendrites[1,3]. Neuronal activation triggers the local translation of these critical mRNAs at synapses allowing spine maturation and elimination, which are essential processes to shape a functional neuronal network in the developing brain. Accordingly, the lack of FMRP expression in FXS patients and *Fmr1* knockout *(Fmr1-KO)* animal models leads to a pathological increase in immature dendritic protrusions due to a failure in the synapse maturation and/or elimination processes[6]. These defects correlate with significant alterations in glutamatergic α-amino-3-hydroxy-5-methyl-4-isoxazolepropionic acid receptor (AMPAR)-mediated synaptic plasticity, including Long-Term Depression (LTD) and Potentiation (LTP)[1,7]. Consequently, these defects lead to learning and memory deficits and underlie the abnormal socio-emotional behaviors in *Fmr1*-KO mice[1,7].

While the CGG-repeat expansion is the most frequent cause of FXS, other mutagenic mechanisms have been reported, including deletions, promoter variants, missense, and nonsense mutations. To date, more than 120 sequence variants have been identified in the *FMR1* gene. However, only three missense mutations (I304N, G266E, and R138Q) have been functionally studied and showed an association with the etiology of FXS[8–13]. Among them, the R138Q mutation is of particular interest since it has been identified in three unrelated individuals presenting clinical traits of FXS. The first male patient sequenced displayed ID, anxiety, and seizures[11,13], while the second presented the classical features of FXS including ID, ADHD, seizures, and ASD[14]. Interestingly, a female with mild ID and attention deficits was recently identified bearing the R138Q mutation[15].

The R138Q mutation does not affect the ability of FMRP to bind polyribosomes and repress the translation of specific target mRNAs[13]. In addition, the intracellular perfusion of a short N-terminal version of FMRP-R138Q in *Fmr1*-KO CA3 hippocampal neurons failed to rescue the action potential broadening, suggesting a functional alteration of the FMRP-R138Q truncated mutant form[13]. However, the cellular and network alterations underlying the phenotype described in FMRP-R138Q FXS patients remain to be elucidated. Here, we have engineered a knock-in mouse model expressing the recurrent missense R138Q mutation in FMRP (*Fmr1^{R138Q}*). We demonstrate that *Fmr1^{R138Q}* mice exhibit postsynaptic alterations in the hippocampus, including an increased dendritic spine density, AMPAR-mediated synaptic plasticity defects associated with severe impairments in their socio-cognitive performances.

## Results

### *Fmr1* mRNA and FMRP protein levels in *Fmr1^{R138Q}* mice.

To assess the pathophysiological impact of the recurrent R138Q mutation in vivo, we generated a specific knock-in mouse line expressing the R138Q FXS mutation using classical homologous recombination in murine C57BL/6 embryonic stem (ES) cells (Fig. 1a). The R138Q coding mutation was introduced in exon 5 by changing the CGA arginine codon into a CAA nucleotide triplet coding for a glutamine (R138Q: c.413G > A). The integrity of the FXS mutation in the *Fmr1^{R138Q}* mice was confirmed by genomic DNA sequencing (Fig. 1a). *Fmr1^{R138Q}* mice were viable, showed a standard growth and normal fertility and mortality rates (Supplementary Fig. 1).

The expression pattern of the FMRP protein is developmentally regulated[16]. To assess whether the R138Q mutation affects the developmental profile of FMRP, we compared the FMRP protein levels in the brain of WT and *Fmr1^{R138Q}* mice at different postnatal days (PND) (Fig. 1b). As expected from the literature[2,5], the FMRP protein level peaked at PND10–15 (WT PND15: 1.330 ± 0.246) and then significantly decreased in the adult brain of WT mice (WT PND90: 0.441 ± 0.078). FMRP-R138Q protein levels showed a similar pattern in the developing *Fmr1^{R138Q}* brain (Fig. 1b; *Fmr1^{R138Q}* PND15: 1.300 ± 0.303; *Fmr1^{R138Q}* PND90: 0.491 ± 0.072), indicating that the R138Q mutation does not alter the protein expression of the pathogenic FMRP.

Since FMRP is an RNA-binding protein regulating the local translation of a large number of mRNAs important to the synaptic function, we compared the total levels of a subset of its target mRNAs in PND90 WT and *Fmr1^{R138Q}* male littermate brains by RT-qPCR (Fig. 1c). We found no significant differences in the total mRNA levels of the FMRP targets tested.

To determine whether the R138Q mutation alters the brain morphology, we next performed a Nissl staining on 20-μm-thick brain coronal slices from PND90 WT and *Fmr1^{R138Q}* male littermates. There were no apparent macroscopic defects in the *Fmr1^{R138Q}* brain and the structural organization of the hippocampus was preserved (Fig. 1d).

### *Fmr1^{R138Q}* mice show an increase in hippocampal spine density.

FMRP is essential to proper spine elimination and maturation[3]. A hallmark of the classical FXS phenotype is a pathological excess of long thin immature dendritic protrusions[17], resulting from a failure in postsynaptic maturation and/or elimination processes. To understand if the R138Q mutation impacts spine maturation and/or elimination, we analyzed the morphology and density of dendritic spines in the *Fmr1^{R138Q}* hippocampus (Fig. 2). We first used attenuated Sindbis viral particles in WT and *Fmr1^{R138Q}* cultured hippocampal neurons at 13 days in vitro (13 DIV) to express free GFP and outline the morphology of dendritic spines[18,19]. We then compared the density and morphology of dendritic spines 20 h post transduction (Fig. 2a). Interestingly, while the length of dendritic spines was similar for both genotypes (WT: 1.563 ± 0.0717 μm; *Fmr1^{R138Q}*: 1.521 ± 0.06193 μm), *Fmr1^{R138Q}* neurons displayed a significant increase in spine density compared to WT neurons (WT: 6.06 ± 0.126 spines per 10 μm; *Fmr1^{R138Q}*: 8.16 ± 0.207 spines per 10 μm).

We also evaluated the characteristics of dendritic spines in the CA1 region of the hippocampus of PND90 WT and *Fmr1^{R138Q}* male littermates using Golgi-Cox staining (Fig. 2b; Supplementary Fig. 2). While there was no difference in spine length and width, *Fmr1^{R138Q}* hippocampal neurons displayed a significant increase in spine density both in basal (Fig. 2b; WT: 8.096 ± 0.232 spines per 10 μm; *Fmr1^{R138Q}*: 10.62 ± 0.167 spines per 10 μm) and apical dendrites (Supplementary Fig. 2; WT: 8.996 ±

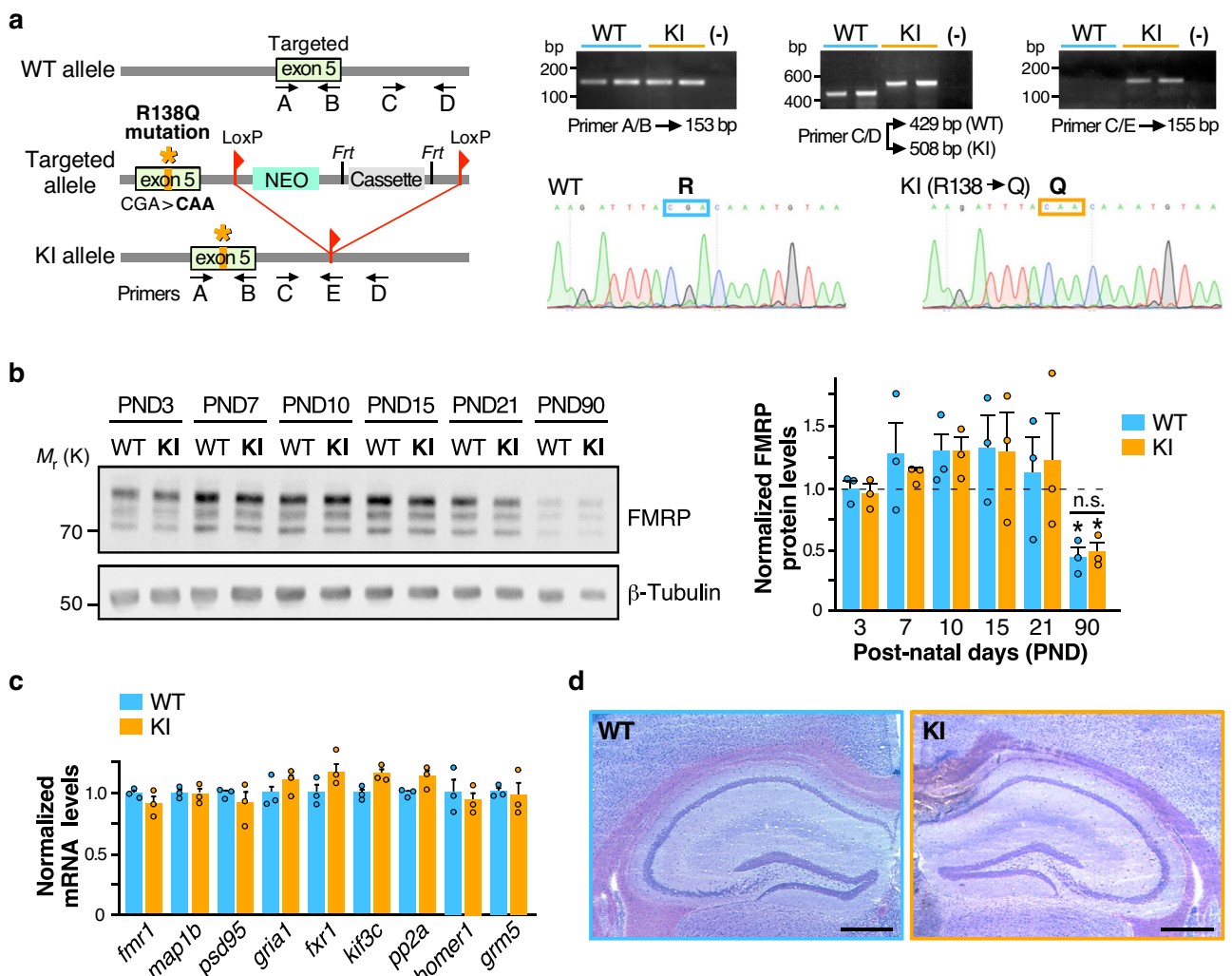

**Fig. 1 Generation and characterization of the *Fmr1^R138Q* Knock-in (KI) mouse line. a** Schematic representation of the *Fmr1* WT allele, the targeting vector used and the allele carrying the missense FXS R138Q mutation. Representative PCR profiles and DNA sequences obtained upon genotyping and genomic DNA sequencing of WT and *Fmr1^R138Q* littermates. **b** Immunoblots showing FMRP protein levels at the indicated postnatal days (PND) in WT and *Fmr1^R138Q* mice. β3-tubulin loading control is also shown. Data are presented as mean values ± s.e.m. of normalized FMRP-WT and FMRP-R138Q protein levels in developing brains of age-matched littermate animals. $N = 3$ biologically independent experiments. Statistical significance determined by Two-way analysis of variance (ANOVA) with Sidak's post test; $*p = 0.0219$ versus PND3–21; ns not significant. **c** Relative abundance of several mRNA targets of FMRP measured by qPCR in PND90 WT and *Fmr1^R138Q* brains. Data are presented as mean values ± s.e.m. of three biologically independent experiments. No significant differences were observed between the genotypes. **d** Representative images of the hippocampal formation in PND90 WT and *Fmr1^R138Q* littermates. Scale bar, 300 μm. Source data are provided as a Source Data file.

0.189 spines per 10 μm; *Fmr1^R138Q*: 9.96 ± 0.152 spines per 10 μm). Taken together, these data indicate that the R138Q mutation rather impairs the elimination of dendritic spines than their maturation given that dendritic spines are morphologically similar in WT and *Fmr1^R138Q* brains.

To go deeper into the characterization of dendritic spines, we performed ultrastructural analyses of WT and *Fmr1^R138Q* hippocampi using transmission electron microscopy (TEM). Stereological analyses pointed out a significant increase in the density of excitatory synapses in the *Fmr1^R138Q* hippocampus (Fig. 2c; WT: 1.371 ± 0.129 synapses per μm³; *Fmr1^R138Q*: 2.187 ± 0.161 synapses per μm³) in agreement with the Golgi-Cox staining data (Fig. 2b, Supplementary Fig. 2). Interestingly, while there was no difference in the length of the postsynaptic densities (PSD; WT: 232.63 ± 6.26 nm; *Fmr1^R138Q*: 222.73 ± 5.64 nm), the PSD thickness in *Fmr1^R138Q* hippocampal neurons was largely reduced (Fig. 2c; WT: 44.70 ± 0.99 nm; *Fmr1^R138Q*: 34.96 ± 0.76 nm). We also measured a significant increase in the density of

synaptic vesicles in *Fmr1^R138Q* hippocampal presynaptic termini (Fig. 2c; WT: 120.48 ± 4.65 vesicles per μm²; *Fmr1^R138Q*: 148.97 ± 4.25 vesicles per μm²). Altogether, these data reveal that the R138Q mutation leads to ultrastructural alterations both in the pre- and postsynaptic compartments.

**Increased surface-expressed AMPAR levels in *Fmr1^R138Q* mice.** To characterize the effect of the R138Q mutation on the composition of synapses, we compared the total protein levels of several pre- and postsynaptic proteins in brain homogenates prepared from WT and *Fmr1^R138Q* male littermates (Fig. 3a; Supplementary Fig. 3). Interestingly, we measured a significant increase in the total amount of the GluA1 AMPAR subunit in the *Fmr1^R138Q* brain (GluA1 *Fmr1^R138Q*: 1.45 ± 0.12 vs WT). All the other proteins investigated in the *Fmr1^R138Q* brain, including proteins involved in the trafficking of AMPAR, showed levels similar to their WT littermates (Fig. 3a; GluA2 *Fmr1^R138Q*: 1.16 ± 0.18; PSD95 *Fmr1^R138Q*: 1.13 ± 0.12; GRIP1 *Fmr1^R138Q*: 1.27 ±

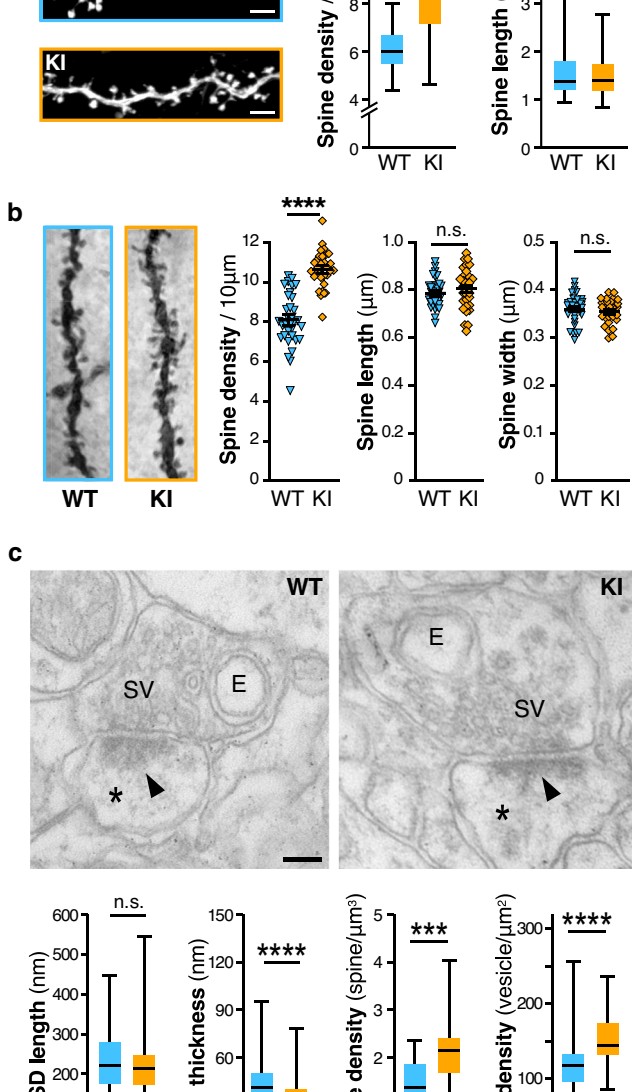

**Fig. 2 The *Fmr1^R138Q* hippocampus exhibits increased dendritic spine density and ultrastructural alterations.** **a** Confocal images of secondary dendrites from GFP-expressing WT and *Fmr1^R138Q* KI cultured hippocampal neurons. Scale bar, 5 μm. Box plots indicate median (middle line), 25th, 75th percentile (box), and min to max values (whiskers) obtained for spine density and length in WT and *Fmr1^R138Q* neurons. N = 48–54 neurons for ~1400–2200 spines analyzed per genotype from six biologically independent experiments. Two-tailed Mann–Whitney test. ****p < 0.0001. **b** Representative images of Golgi-stained basal secondary dendrites of CA1 hippocampal neurons from PND90 WT and *Fmr1^R138Q* littermates. Histograms show the density of spines, spine length and width from WT and *Fmr1^R138Q* CA1 secondary dendrites. Error bars represent the mean ± s. e.m. N = 30 neurons per genotype from three biologically independent experiments (1500–2000 spines analyzed per genotype). Two-sided Mann–Whitney test; ****p < 0.0001. **c** Representative EM images of pre- and postsynaptic (*) elements in CA1 synapses of PND90 WT and *Fmr1^R138Q* hippocampi. E endosomes, SV synaptic vesicles, Arrowheads, postsynaptic densities (PSD). Scale bar, 100 nm. Box plots indicate median (line), 25th, 75th percentile (box), and min to max values (whiskers) for PSD length and thickness, the density of synapses and synaptic vesicles in WT and *Fmr1^R138Q* CA1 hippocampal neurons. Approximately 130 PSD (length and thickness), 60 presynaptic boutons, and 350 μm² of total surface area (synapse density) per genotype were analyzed from three independent sets of the experiment. Unpaired *t* test. ns not significant. ***p = 0.0003; ****p < 0.0001. Source data are provided as a Source Data file.

(Fig. 3a), the surface expression of GluA2 was significantly higher in cultured *Fmr1^R138Q* hippocampal neurons (Fig. 3c; GluA2 *Fmr1^R138Q*: 1.50 ± 0.09 vs WT).

We then examined the surface expression of AMPARs in acute hippocampal slices using BS3-crosslinking assays (Fig. 3d). Consistent with the above data, the surface expression of both GluA1 and GluA2 was also significantly increased in the *Fmr1^R138Q* hippocampus (Fig. 3d, Lanes +BS3; GluA1 *Fmr1^R138Q*: 1.76 ± 0.21 vs WT; Lanes +BS3; GluA2 *Fmr1^R138Q*: 1.58 ± 0.17 vs WT). Altogether, these data clearly indicate that the R138Q mutation leads to increased surface levels of both GluA1 and GluA2 in vitro and in vivo.

**Altered synaptic transmission in the *Fmr1^R138Q* hippocampus.** We showed that the recurrent FXS R138Q missense mutation leads to an increase in AMPAR surface expression. To assess whether this increase occurs at least in part synaptically, we performed super-resolution STimulated Emission Depletion (STED) microscopy on surface-labeled GluA1 or GluA2 in WT and *Fmr1^R138Q* cultured hippocampal neurons (Fig. 4a–d). We first measured the mean fluorescence intensity from surface-expressed GluA1 and GluA2 at Homer1-labeled postsynaptic sites (Fig. 4a-c). We found that the mean surface GluA1 fluorescence intensity per synapse is significantly increased in *Fmr1^R138Q* neurons (Fig. 4c, WT: 797 ± 28; *Fmr1^R138Q*: 1013 ± 34), whereas the fluorescence associated with surface-expressed GluA2 is reduced (Fig. 4c, WT: 1373 ± 34; *Fmr1^R138Q*: 1148 ± 29). At the postsynapse, AMPARs are organized in 80–90 nm nanodomains facing presynaptic glutamate release sites for efficient synaptic transmission[20]. Thus, we compared the mean number of nanodomains containing surface-expressed GluA1 and GluA2 in WT and *Fmr1^R138Q* hippocampal neurons (Fig. 4a, b, d). Consistent with the literature[20], we measured a density of ~2–2.1 nanodomains per spine for both surface-associated GluA1 and GluA2 in WT neurons (Fig. 4d, WT sGluA1: 2.023 ± 0.107; WT sGluA2: 2.121 ± 0.099 nanodomains per spine). Interestingly, in *Fmr1^R138Q* spines there was a significant increase in the mean

0.19; PICK1 *Fmr1^R138Q*: 0.92 ± 0.15; Stargazin *Fmr1^R138Q*: 1.02 ± 0.03).

To further explore the AMPAR defects in the *Fmr1^R138Q* brain, we compared the levels of surface-expressed GluA1 in DIV15 WT and *Fmr1^R138Q* cultured hippocampal neurons (Fig. 3b). Using surface-immunolabeling assays with specific anti-GluA1 antibodies, we showed that the surface levels of GluA1 in *Fmr1^R138Q* neurons were significantly increased (Fig. 3b; Mean surface GluA1 intensity, WT: 1 ± 0.087; *Fmr1^R138Q*: 1.42 ± 0.13), with a higher density of surface GluA1-containing clusters (Fig. 3b; Surface cluster density, WT: 1 ± 0.049; *Fmr1^R138Q*: 1.22 ± 0.06). Since AMPARs are often concentrated in dendritic spines, this finding corroborates the increased number of dendritic spines measured in *Fmr1^R138Q* neurons (Fig. 2). We further confirmed the significant increase in GluA1 surface expression in *Fmr1^R138Q* neurons using cell surface biotinylation assays (Fig. 3c; GluA1 *Fmr1^R138Q*: 1.85 ± 0.19 vs WT). Interestingly, while there was no alteration in the total levels of GluA2 in the *Fmr1^R138Q* brain

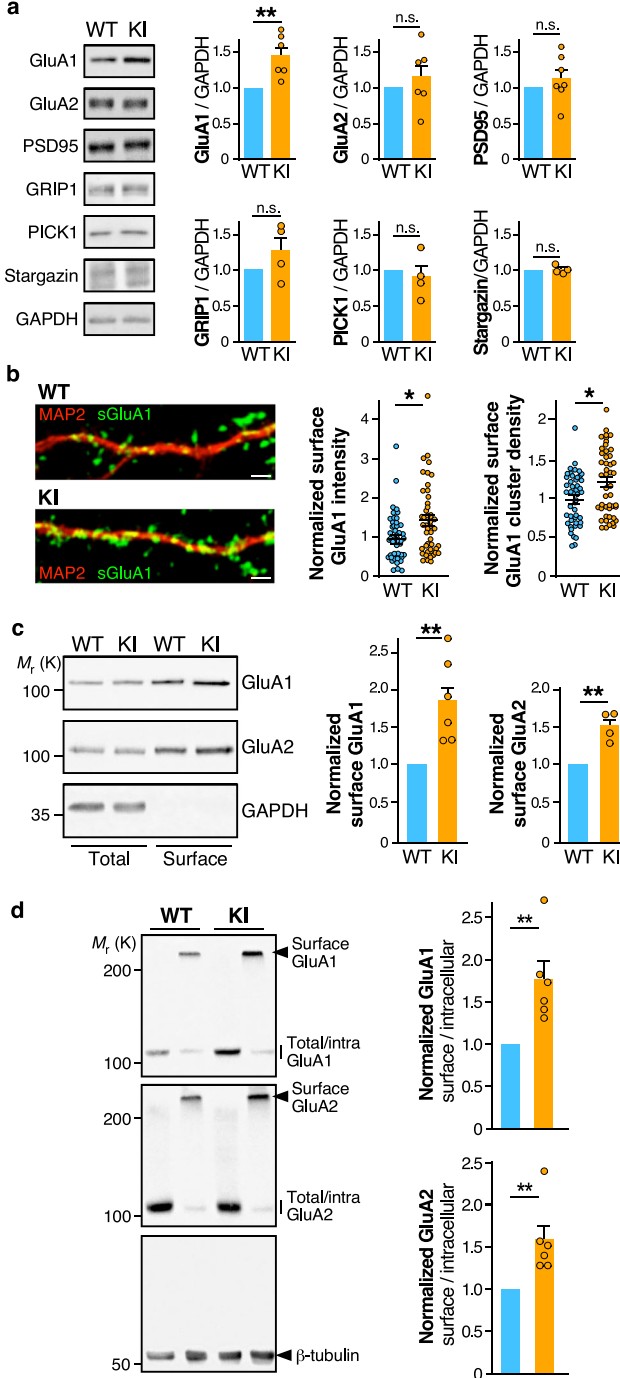

**Fig. 3 Increased surface expression of AMPAR in hippocampal neurons of *Fmr1^R138Q* KI mice. a** Immunoblots showing the levels of the indicated synaptic proteins in brain homogenates from PND21 WT and *Fmr1^R138Q* littermate animals. GAPDH was used as a loading control. Quantification shows the mean ± s.e.m. of the total levels of the indicated proteins. *N* = 6 (GluA1, GluA2), *N* = 7 (PSD95), and *N* = 4 (GRIP1, PICK1, Stargazin) biologically independent experiments. Unpaired *t* test. ns not significant. **\**p* = 0.0034. **b** Secondary dendrites from TTX-treated WT and *Fmr1^R138Q* hippocampal neurons at 15 DIV stained for surface GluA1 (green) and MAP2 (red). Scale bar, 5 μm. Histograms show mean ± s.e.m. of both surface intensity and cluster density for GluA1 in WT and *Fmr1^R138Q* neurons. Values were normalized to their respective basal conditions. *N* = 47 neurons per genotype from five biologically independent experiments. Two-tailed Mann–Whitney test. **p* = 0.0147 (intensity) and **p* = 0.0124 (density). **c** Immunoblots showing the surface expression of GluA1 and GluA2 in TTX-treated WT and *Fmr1^R138Q* cultured hippocampal neurons at 15 DIV using biotinylation assays. Histograms show the mean ± s.e.m. of the normalized level of GluA1 and GluA2 subunits at the neuronal surface in WT and *Fmr1^R138Q* neurons. *N* = 7 (GluA1) and 4 (GluA2) biologically independent experiments respectively. Two-sided ratio *t* test. **\**p* = 0.0011 (GluA1); **\**p* = 0.0074 (GluA2). **d** Immunoblots showing the basal surface expression of GluA1 and GluA2 in PND90 TTX-treated WT and *Fmr1^R138Q* hippocampal slices using the BS3-crosslinking assay. Control tubulin immunoblot is included to confirm the absence of BS3 crosslinking intracellularly. The surface/intracellular ratio in the WT was set to 1 and *Fmr1^R138Q* values were calculated respective to the WT. Error bars show the mean values ± s.e.m. *N* = 6 independent experiments. Two-tailed ratio *t* test. **\**p* = 0.0041 (GluA1); **\**p* = 0.0056 (GluA2). Source data are provided as a Source Data file.

Finally, to better understand whether the increase in surface-expressed AMPARs measured in the *Fmr1^R138Q* hippocampus (Figs. 2, 3, and 4a–d) is associated with alterations in glutamatergic transmission, we performed whole-cell patch-clamp recordings in CA1 neurons from hippocampal slices of PND90 WT and *Fmr1^R138Q* littermates (Fig. 4e–k). We showed that the amplitude of AMPAR-mediated miniature Excitatory PostSynaptic Currents (mEPSCs) is significantly enhanced in *Fmr1^R138Q* mice (Fig. 4e, f; WT: 17.09 ± 0.539 pA; *Fmr1^R138Q*: 18.97 ± 0.405 pA). Interestingly, we did not measure any significant differences in the frequency of mEPSCs (Fig. 4g, h; WT: 0.178 ± 0.040 Hz; *Fmr1^R138Q*: 0.143 ± 0.025 Hz) or the kinetics of these events (Fig. 4i, j) between the two genotypes.

Altogether the data from the above experiments (Figs. 2–4) revealed that the R138Q FXS mutation leads to important pre- and postsynaptic alterations resulting in synaptic transmission deficits in the *Fmr1^R138Q* hippocampus.

**Impaired long-term potentiation in the *Fmr1^R138Q* hippocampus.** We next investigated the consequences of the R138Q mutation in hippocampal plasticity. Since the level of surface-expressed AMPARs is enhanced in the *Fmr1^R138Q* hippocampus, we wondered whether the induction of LTP could trigger a further increase in synaptic AMPARs (Fig. 5 and Supplementary Fig. 4). To test this hypothesis, we first combined surface immunolabeling assays with the chemical induction of LTP (cLTP[21]) on WT and *Fmr1^R138Q* hippocampal neurons (Fig. 5a, b). In line with the literature[21], the level of surface GluA1 was significantly increased upon cLTP in WT neurons (Mean surface GluA1 intensity, WT: 1.427 ± 0.129 vs basal; Surface cluster density, WT: 1.308 ± 0.08596 vs basal) whereas the surface level of GluA1 in *Fmr1^R138Q* neurons was unexpectedly decreased (Mean surface GluA1 intensity, *Fmr1^R138Q*: 0.6354 ± 0.08647 vs basal; Surface cluster density, *Fmr1^R138Q*: 0.7455 ± 0.08227 vs basal).

number of both surface-associated GluA1 and GluA2 nanodomains (Fig. 4d, *Fmr1^R138Q* sGluA1: 2.582 ± 0.106; *Fmr1^R138Q* sGluA2 2.554 ± 0.078). These data indicate that the missense R138Q mutation leads to a significant increase in the number of postsynaptic nanodomains containing AMPARs. However, while the mean fluorescence associated with surface-expressed GluA1 is also increased in hippocampal *Fmr1^R138Q* synapses, the synaptic fluorescence level from surface-labeled GluA2 is decreased revealing that the upregulation of surface GluA2 measured in biochemical experiments is rather due to its extrasynaptic increase. This indicates that the FXS mutation not only impacts the surface expression of both GluA1 and GluA2 but also differentially perturbs their synaptic targeting, trafficking, and nanoscale organization.

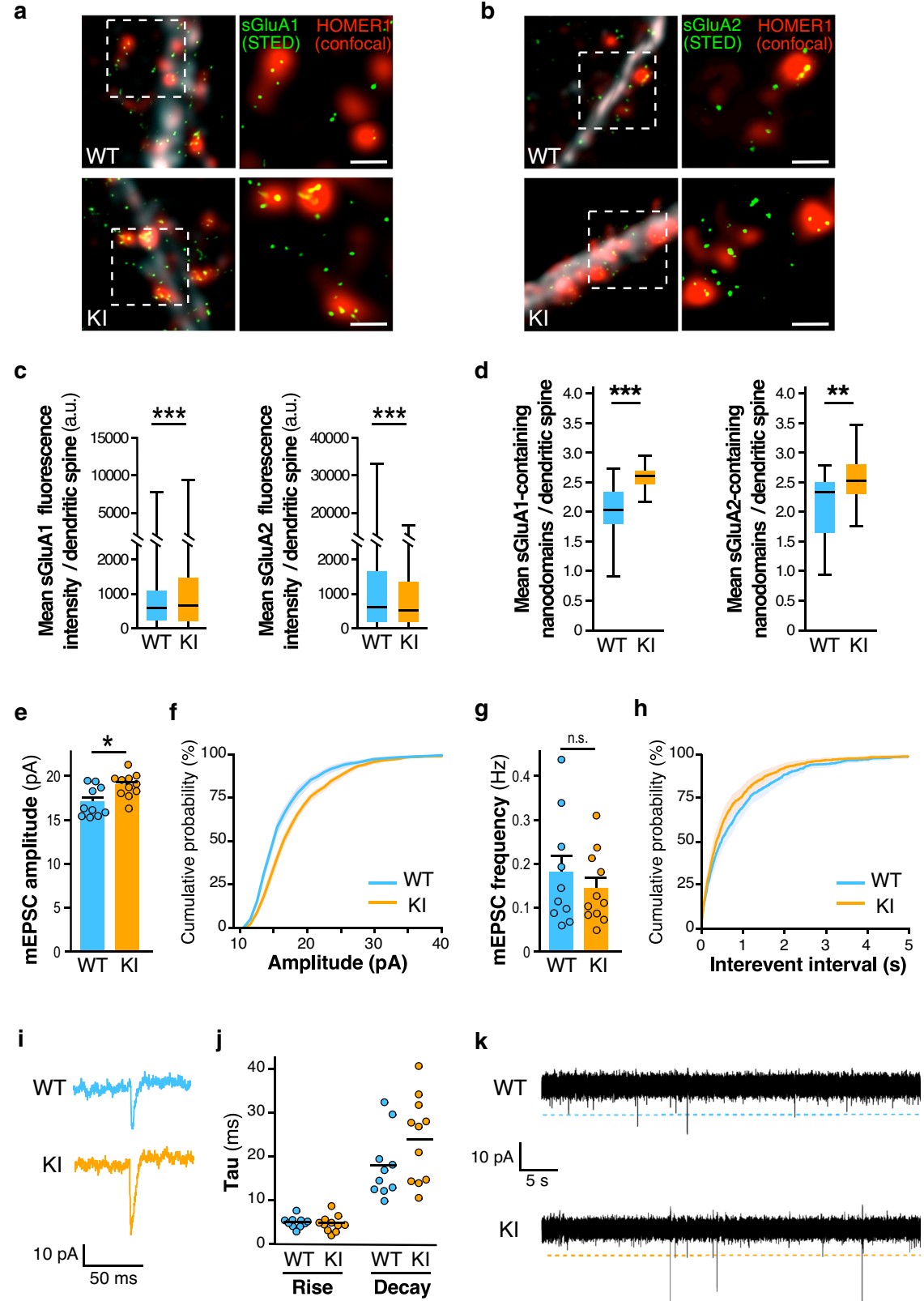

We confirmed these results using surface biotinylation assays in WT and *Fmr1*[R138Q] hippocampal neurons in basal and cLTP-induced conditions (Supplementary Fig. 4a–c). The cLTP treatment triggered the increase in both GluA1 and GluA2 surface expression in WT neurons (WT GluA1 cLTP: 1.578 ± 0.107; WT GluA2 cLTP: 1.203 ± 0.008). In contrast, the

surface levels of AMPARs upon cLTP were not increased but as above, rather decreased in *Fmr1*[R138Q] hippocampal neurons (*Fmr1*[R138Q] GluA1 cLTP: 0.490 ± 0.073; *Fmr1*[R138Q] GluA2 cLTP: 0.899 ± 0.157).

In addition to the data obtained on hippocampal cultures, we performed BS3-crosslinking assays and showed that the induction

**Fig. 4 Increased synaptic surface expression of AMPAR and basal excitatory transmission in hippocampal $Fmr1^{R138Q}$ neurons. a, b** Super-resolution STED images of surface-expressed GluA1 and GluA2 (STED, green) in postsynaptic Homer1 sites (confocal, red) of TTX-treated WT and $Fmr1^{R138Q}$ hippocampal neurons. Scale bar, 500 nm. **c, d** Quantification of (**a**) and (**b**). Box plots indicate median (middle line), 25th, 75th percentile (box), and min to max values (whiskers) obtained for the mean surface GluA1 and GluA2 fluorescence intensity (**c**) and nanodomains (**d**) in WT and $Fmr1^{R138Q}$ neurons. N = 17–30 (WT) and 15–30 ($Fmr1^{R138Q}$) neurons were analyzed from three biologically independent experiments with a total of dendritic spines analyzed ranging from 878 to 4246. Unpaired $t$ test. ***$p < 0.0001$; **$p = 0.0011$. **e–k** mEPSCs recordings from acute hippocampal slices obtained from PND90 WT and $Fmr1^{R138Q}$ littermates. Quantification shows the mean values ± s.e.m. (**e, g**) and cumulative curves ± s.e.m. (**f, h**) of mEPSC amplitude and frequency. N = 10–11 neurons per genotype from three independent experiments. Unpaired $t$ test. ns not significant. *$p = 0.0111$. **i** Example traces of WT and $Fmr1^{R138Q}$ single events. **j** Computed Tau rise and decay. **k** Representative traces of mEPSC recordings from WT and $Fmr1^{R138Q}$ hippocampal slices. Source data are provided as a Source Data file.

of cLTP in $Fmr1^{R138Q}$ hippocampal slices led to a significant reduction in the levels of surface-expressed AMPARs (Fig. 5c–e, $Fmr1^{R138Q}$ GluA1 cLTP: $0.706 \pm 0.008$ vs basal; Supplementary Fig. 4d–f, $Fmr1^{R138Q}$ GluA2 cLTP: $0.860 \pm 0.05$ vs $Fmr1^{R138Q}$ basal), further confirming that the R138Q mutation impairs the AMPAR trafficking. As expected, the cLTP treatment was able to increase the surface levels of AMPARs in hippocampal slices from WT littermate brains (Fig. 5c, e, WT GluA1 cLTP: $3.691 \pm 0.675$ vs basal; Supplementary Fig. 4d–f, WT GluA2 cLTP: $1.953 \pm 0.315$ vs WT basal).

Altogether, these data indicate that the hippocampal plasticity is severely impaired in the $Fmr1^{R138Q}$ mice.

It has been demonstrated that the LTP induction promotes a reorganization of the AMPAR nanodomains[22]. Thus, to further assess the impact of the R138Q mutation in cLTP-activated neurons, we performed super-resolution STED microscopy on surface-labeled GluA1 in cLTP-treated WT and $Fmr1^{R138Q}$ hippocampal cells (Fig. 5f–h). As expected, the mean fluorescence intensity per synapse (Fig. 5f, g; WT basal: $797 \pm 28$; WT cLTP: $1076 \pm 35$), as well as the density of the postsynaptic nanodomains (Fig. 5f, h; WT basal: $2.023 \pm 0.107$; WT cLTP: $2.783 \pm 0.09$ nanodomains per spine) containing the surface-expressed GluA1 subunits were both significantly increased in WT cLTP-treated neurons. In contrast, $Fmr1^{R138Q}$ neurons exhibited a significant decrease in the mean postsynaptic surface GluA1 fluorescence intensity upon cLTP (Fig. 5f, g; $Fmr1^{R138Q}$ basal: $1013 \pm 34$; $Fmr1^{R138Q}$ cLTP: $890 \pm 30$) while the mean density of surface GluA1-containing nanodomains remained unchanged (Fig. 5f, h; $Fmr1^{R138Q}$ sGluA1 basal: $2.582 \pm 0.05$; $Fmr1^{R138Q}$ sGluA1 cLTP: $2.599 \pm 0.040$ nanodomains per spine). These data thus reveal that the R138Q mutation also impacts the synaptic reorganization of AMPARs upon cLTP in $Fmr1^{R138Q}$ cultured hippocampal neurons, further confirming that the activity-dependent trafficking of these receptors is impaired in the $Fmr1^{R138Q}$ mice.

Altogether, these data indicate that, contrary to WT neurons, the induction of cLTP does not promote any increase in the surface nor the synaptic levels of AMPARs in the $Fmr1^{R138Q}$ hippocampus. Therefore, to determine if the expression of the FMRP-R138Q mutant also physiologically impacts the AMPAR-mediated responses, we induced LTP by high-frequency stimulation (HFS) in acute WT and $Fmr1^{R138Q}$ hippocampal slices and recorded the postsynaptic responses in CA1 neurons (Fig. 5i–k). First, we tested the impact of the R138Q mutation on the CA3 to CA1 synaptic transmission and did not find any significant differences with the WT responses in Input/Output curves established following the stimulation of the Schaffer collaterals (Supplementary Fig. 5). This indicates that the connectivity between the pre- and postsynaptic sites is preserved in the $Fmr1^{R138Q}$ hippocampus. Next, we found that the induction of LTP by HFS was evoked as expected in the WT hippocampus (WT LTP: $157.2 \pm 6.52\%$ vs basal) but was drastically reduced in $Fmr1^{R138Q}$ male littermates ($Fmr1^{R138Q}$

LTP: $122.4 \pm 10.12\%$ vs basal), in line with the impaired cLTP seen in both biochemical and imaging experiments (Fig. 5a–h). In addition, we did not measure any significant difference in the mean fiber volley (FV) slope between genotypes (Fig. 5i-k), suggesting that the impaired LTP in the $Fmr1^{R138Q}$ hippocampus rather arises from postsynaptic impairments than from presynaptic alterations.

**$Fmr1^{R138Q}$ mice display ID- and ASD-like features**. The R138Q mutation has been identified in both male and female patients[11,13,15]. Since the mutation affects the surface levels of AMPARs and directly impacts synaptic plasticity in the hippocampus, we investigated whether male and female $Fmr1^{R138Q}$ mice display altered cognitive and/or social performances (Fig. 6). To avoid possible pitfalls due to an impact of the R138Q mutation on locomotion, we first tested PND40–45 mice using the open field test to detect any potential motor defects. We did not measure any significant differences in the number of crossings between the two genotypes demonstrating that there is no motor alteration in the $Fmr1^{R138Q}$ mice (Supplementary Fig. 6).

Communication skills are altered in some FXS patients and $Fmr1$-KO mice[19]. We therefore compared the ultrasonic vocalization (USV) profile in PND7 WT and $Fmr1^{R138Q}$ pups removed from the nest and found a ~50% decrease in the number of USVs in $Fmr1^{R138Q}$ compared to WT animals in both genders (Fig. 6a, b; $Fmr1^{R138Q}$ male: $75.36 \pm 17.71$ USVs; WT male: $147.4 \pm 14.06$ USVs; $Fmr1^{R138Q}$ female: $79.88 \pm 35.68$ USVs; WT female: $163.4 \pm 31.49$ USVs). These data indicate that the R138Q mutation leads to severe communicative deficits in infant $Fmr1^{R138Q}$ mice.

Next, we evaluated the cognitive performance in PND40–45 males and females with the novel object recognition test (Fig. 6c, d). We found a profound deficit for both genders as the $Fmr1^{R138Q}$ mice spent significantly less time than WT animals exploring the novel object ($Fmr1^{R138Q}$ male: $51.18 \pm 3.26\%$; WT male: $71.16 \pm 3.72\%$; $Fmr1^{R138Q}$ female: $45.27 \pm 4.37\%$; WT female: $66.42 \pm 3.31\%$). In addition, the $Fmr1^{R138Q}$ mice were spending significantly more time sniffing the old object ($Fmr1^{R138Q}$ male: $27.27 \pm 3.85$ s; WT male: $16.46 \pm 2.46$ s; $Fmr1^{R138Q}$ female: $19.22 \pm 3.13$ s; WT female: $9.77 \pm 0.98$ s), thus showing a lower discrimination index (Discrimination index $Fmr1^{R138Q}$ male: $2.36 \pm 6.51\%$; WT male: $42.31 \pm 7.45\%$; Discrimination index $Fmr1^{R138Q}$ female: $-9.45 \pm 8.73\%$; WT female: $32.84 \pm 6.63\%$). These data indicate that the R138Q mutation substantially alters the cognitive function in the $Fmr1^{R138Q}$ mice.

About 25% of FXS patients present ASD traits, including social avoidance and decreased social skills. Therefore, to evaluate the impact of the R138Q mutation in ASD-like behaviors, we tested sociability in both male and female WT and $Fmr1^{R138Q}$ mice using the three-chamber test (Fig. 6e, f). As expected, WT animals from both sexes spent significantly more time sniffing the novel mouse rather than the empty cage. We showed that both the time spent sniffing the social stimulus and the ability to discriminate

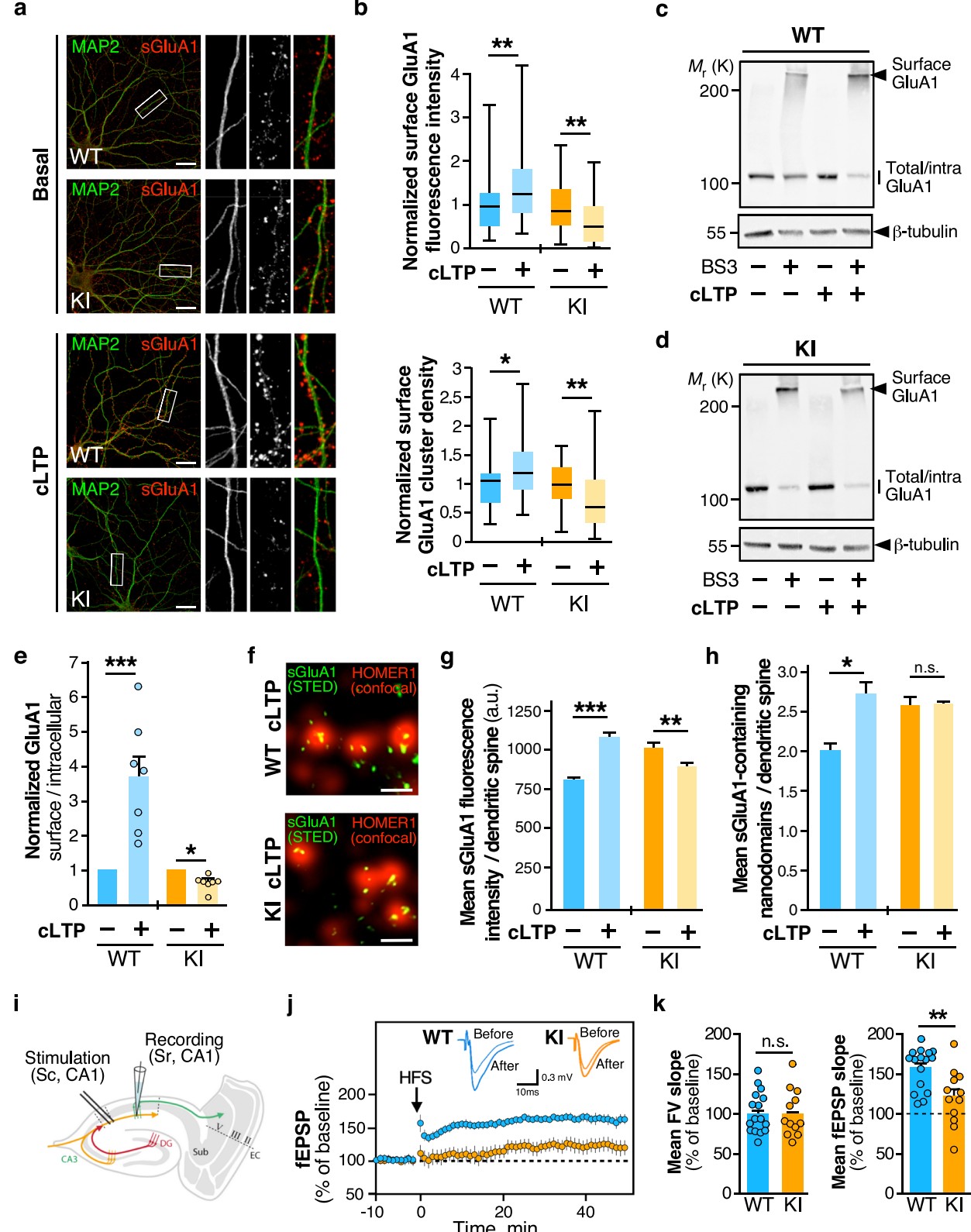

between the novel mouse and the empty cage was dramatically reduced in *Fmr1*[R138Q] males (*Fmr1*[R138Q] male sniffing time: 42.45 ± 13.84%; WT male: 80.48 ± 4.39%; Discrimination index *Fmr1*[R138Q] male: −15.10 ± 27.69%; WT male: 60.96 ± 8.78%). However, there was no significant difference between the WT and *Fmr1*[R138Q] female mice (*Fmr1*[R138Q] female sniffing time: 63.21 ±

9.25%; WT female: 69.29 ± 10.69%; Discrimination index *Fmr1*[R138Q] female: 26.41 ± 18.51%; WT female: 38.58 ± 21.37%).

Altogether, our behavioral data are in line with our biochemical and physiological data and reveal that *Fmr1*[R138Q] mice display important communicative, cognitive, and social deficits.

**Fig. 5 LTP is impaired in *Fmr1^R138Q* mice. a** Secondary dendrites from TTX-treated WT and *Fmr1^R138Q* 15 DIV neurons stained for MAP2-positive microtubule (green) and surface-expressed GluA1 (red) in basal conditions and upon cLTP induction. Bar, 20 μm. **b** Boxplots indicate median (line), 25^th, 75^th percentile (box), and min-to-max values (whiskers) obtained for surface GluA1 intensity and cluster density in WT and *Fmr1^R138Q* neurons in control and cLTP conditions. Values were normalized to their basal conditions. $N = 39–42$ neurons per genotype from four independent experiments. Two-tailed Mann–Whitney test. $*p = 0.0103$; $**p = 0.0053$ (WT sGluA1 intensity); $**p = 0.0034$ (KI sGluA1 intensity); $**p = 0.0036$ (KI sGluA1 cluster). **c, d** Immunoblots showing the surface expression of GluA1 in basal and cLTP-induced conditions in PND90 TTX-treated WT (**c**) and *Fmr1^R138Q* (**d**) hippocampal slices using BS3-crosslinking assays. Control Tubulin immunoblot is included to control the absence of intracellular BS3-crosslinking. **e** The surface/intracellular ratio in the WT was set to 1 and *Fmr1^R138Q* values were calculated respective to the WT. Bars show the mean ± s.e.m. $N = 7$ independent experiments. Two-tailed ratio t test. $*p = 0.0358$; $***p = 0.0004$. **f** STED images of surface-expressed GluA1 (STED, green) in postsynaptic Homer1 sites (confocal, red) of cLTP induced WT and *Fmr1^R138Q* hippocampal neurons. Scale bar, 500 nm. **g, h** Box plots indicate median (middle line), 25^th, 75^th percentile (box), and min to max values (whiskers) obtained for the postsynaptic surface-associated GluA1 fluorescent intensity (**g**) and nanocluster density (**h**) computed from STED imaging data in basal and cLTP-treated WT and *Fmr1^R138Q* neurons. $N = 13–17$ (WT) and 15–18 (*Fmr1^R138Q*) neurons were analyzed from three independent experiments. **g, h** Unpaired t test. $***p < 0.0001$; $**p = 0.0079$. ns not significant. Values for control surface GluA1 in (**f**–**h**) are taken from Fig. 4 (**c, d**) since these experiments were performed in parallel. **i** Schematic diagram of the stimulating and recording areas in the mouse hippocampus. **j** fEPSPs were recorded at CA1 synapses on hippocampal slices from P35-42 WT and *Fmr1^R138Q* littermates in basal conditions and upon LTP induction by high-frequency stimulation (HFS, 3 x 100Hz, 1 s). **k** Histograms show the mean ± s.e.m. of fiber volley (FV) and fEPSP slopes from 12–16 neurons per genotype in four independent experiments. Unpaired t test with Welch's correction. ns not significant; $**p = 0.0092$. Source data are provided as a Source Data file.

## Discussion

Synaptic transmission and/or plasticity defects have been clearly linked to the development of many, if not all, neurological disorders. Therefore, a better understanding of the pathways underlying these alterations is essential to develop strategies to rescue the identified dysfunctions and design innovative targeted therapies to treat these diseases. Here, we generated and characterized a novel mouse model for FXS expressing the recurrent R138Q missense mutation in the FMRP protein. We show that the R138Q mutation leads to an increase in spine density, alterations in both the pre- and postsynaptic organization, and an impaired LTP in the *Fmr1^R138Q* hippocampus. The consequence of this plasticity defect is an abnormal socio-cognitive behavior in *Fmr1^R138Q* mice that resembles the ID and ASD-like traits described in FXS patients bearing the R138Q mutation. Altogether, our data validate the *Fmr1^R138Q* mouse line as a compelling preclinical model to investigate the molecular mechanisms underlying the pathology.

To date, only two studies have provided some insights into the impact of the R138Q mutation on neuronal function[13,23]. FMRP is known to participate in the regulation of AMPAR trafficking[1,7], which is critical to maintaining the synaptic function. Alpatov and colleagues[23] investigated the impact of the R138Q mutation on the basal trafficking of AMPAR and reported that the exogenous expression of FMRP-R138Q does not impact the constitutive endocytosis of AMPAR in an *Fmr1-KO* background. Here, we revealed an altered surface expression of both GluA1 and GluA2 AMPAR subunits within the postsynaptic membrane of the *Fmr1^R138Q* hippocampus, participating in an LTP impairment in these mice. We also identified that the R138Q mutation leads to alterations in the postsynaptic organization of AMPAR-containing nanodomains (Fig. 4). Nanoscale scaffolding domains at the PSD are essential to organize and concentrate AMPARs to allow an efficient synaptic transmission[20,24]. These 80-nm AMPAR nanodomains facing presynaptic glutamate release sites are dynamic and cLTP induction promotes their nanoscale reorganization[20,22,24]. Importantly, the calcium-dependent function of AMPARs also relies on the presence of edited GluA2 subunits at the synapse, which confers a low $Ca^{2+}$ permeability to GluA2-containing heteromers (calcium impermeable, CI-AMPARs). Conversely, GluA1 homomers, which lack the GluA2 subunit, are permeable to $Ca^{2+}$ and refer as CP-AMPARs[25–27]. In basal conditions, the majority of AMPAR assemblies in the adult hippocampus contain edited $Ca^{2+}$-impermeable GluA2 subunits, which is essential to maintaining

a low intracellular $Ca^{2+}$ concentration. In the *Fmr1^R138Q* hippocampal synapses, there is an excess of GluA1 subunits associated with a decrease in the overall amount of surface-expressed GluA2 at the synapse (Fig. 4a–d). This likely reflects a different AMPAR subunit composition in *Fmr1^R138Q* synapses, potentially favoring the formation of CP-AMPARs. The recruitment of CP-AMPARs to postsynaptic sites plays a key role in the initial phase of LTP[22,25,26]. Interestingly, while we measured an overall synaptic increase in CP-AMPARs upon cLTP induction in WT neurons, this event is compromised in the *Fmr1^R138Q* hippocampus (Fig. 5g). Thus, it is tempting to speculate that the enhanced expression of synaptic CP-AMPARs in basal conditions in *Fmr1^R138Q* neurons likely impairs the recruitment of additional GluA1-containing CP-AMPARs that are necessary to initiate the LTP. Furthermore, we showed here that, contrary to the WT synapse, there is no increase in GluA1-containing nanodomains upon LTP in *Fmr1^R138Q* neurons (Fig. 5h), which could also participate in the impaired postsynaptic response measured in the *Fmr1^R138Q* hippocampus.

The *Fmr1^R138Q* postsynaptic compartment also presents important ultrastructural alterations and displays an increase in the density of presynaptic neurotransmitter vesicles and a significant reduction of the PSD thickness (Fig. 2). This could lead to defective synaptic responses including altered diffusive properties and/or anchoring of AMPARs within the postsynaptic membranes, which is critical to recruiting AMPARs at the hippocampal synapse upon LTP induction[28,29].

Altogether, these data uncover a previously unsuspected postsynaptic impact of the R138Q mutation leading to both basal and activity-dependent AMPAR trafficking defects in the *Fmr1^R138Q* hippocampus.

The second study investigating the functional impact of the R138Q mutation reported that the exogenous overexpression of a truncated (FMRP_{1–298}) version of the FMRP-R138Q mutant fails to rescue action potential (AP) broadening in *Fmr1-KO* neurons[13], which correlates with an increased presynaptic release[30]. However, the presynaptic release per se was not assessed in this work. The authors also showed that the R138Q mutation disrupts the interaction of the short FMRP_{1–298} form with the β4 subunit of BK channels, thus underlying AP duration[13,30]. To conclude, they hypothesized that an alteration of the presynaptic function is likely responsible for the ID and seizures exhibited by the first FXS R138Q patient[13]. Importantly, BK channels are localized both at pre- and postsynaptic sites[31]. Consequently, alterations in FMRP/BK channel interaction may not only be linked to a

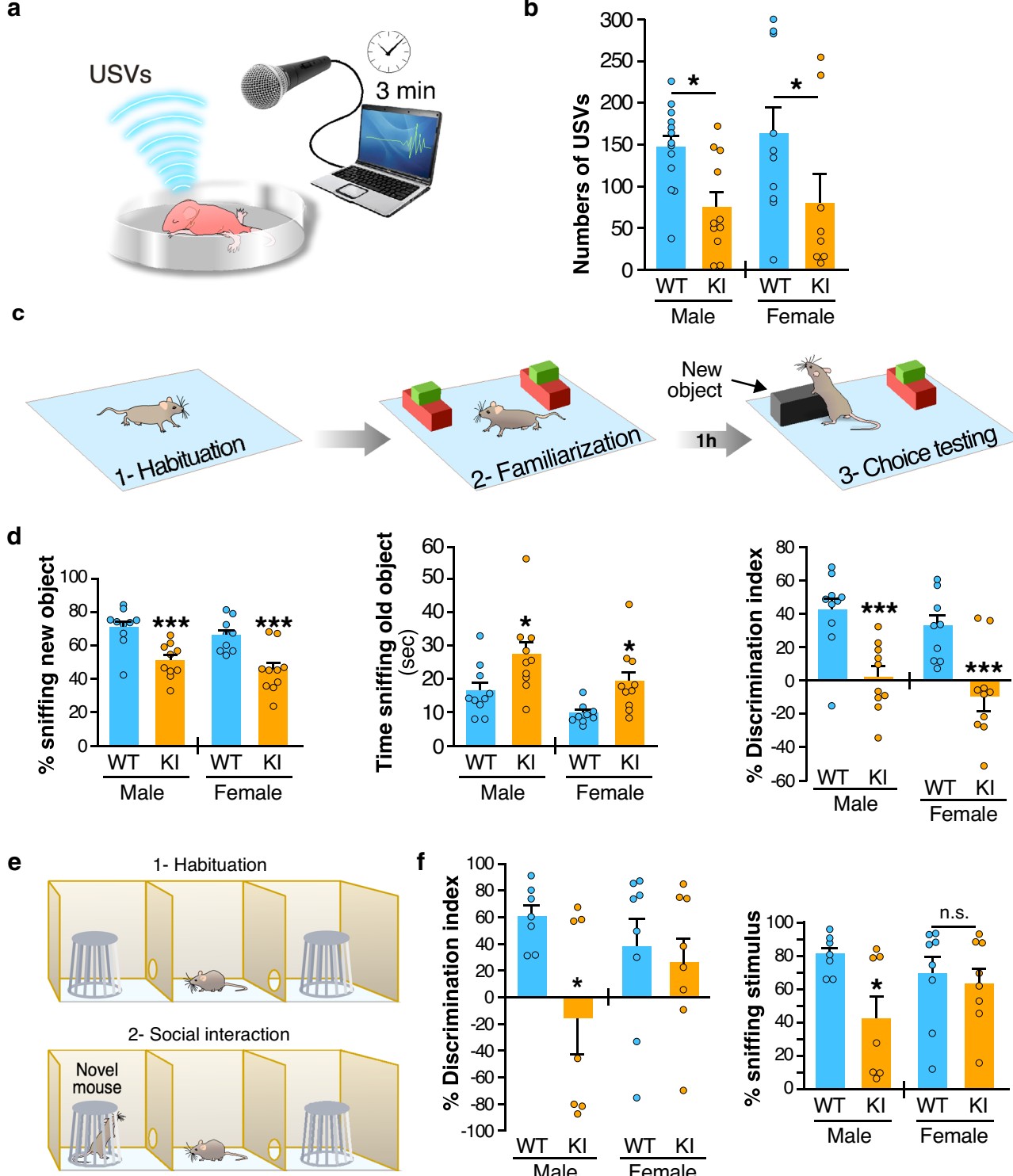

**Fig. 6 *Fmr1^{R138Q}* mice show communication deficits and socio-cognitive alterations. a** Schematic of the isolation-induced ultrasonic vocalizations (USVs) test. **b** Compared to WT animals, *Fmr1^{R138Q}* KI mice (PND7) emit ~50% less USVs when removed from the nest. Histogram shows the mean values ± s.e.m. of USVs in WT and *Fmr1^{R138Q}* males and females. WT, $N = 13$ males, 11 females; *Fmr1^{R138Q}*, $N = 10$ males, 8 females. *$p = 0.028$. **c** Schematic of the object recognition test used to assess the cognitive domain. **d** Quantification shows the mean values ± s.e.m. of time sniffing the new object (%), the old object (s), and the discrimination index (%) for both PND40–45 WT and *Fmr1^{R138Q}* males and females. WT, $N = 10$ males, 9 females; *Fmr1^{R138Q}*, $N = 10$ males, 10 females. *$p = 0.011$ (male), *$p = 0.028$ (female), ***$p < 0.001$. **e** Scheme of the three-chamber test used to assess sociability. **f** Histograms show the mean percentage ± s.e.m. of discrimination index and time sniffing the stimulus mouse for both PND40–45 WT and *Fmr1^{R138Q}* males and females. WT, $N = 7$ males, 8 females; *Fmr1^{R138Q}*, $N = 7$ males, 8 females. *$p = 0.017$. Two-way ANOVA with genotype and sex as factors followed by Newman–Keuls post-hoc test for individual group comparisons were computed for all behavioral studies. Source data are provided as a Source Data file.

presynaptic impairment as suggested by the authors but also to the postsynaptic defects unraveled in the present work.

In the present study, we also demonstrated that the R138Q mutation leads to an increase in the density of synapses in the $Fmr1^{R138Q}$ hippocampus. Unexpectedly, the frequency of mEPSCs, which usually correlates with synaptic density, is not altered (Fig. 4e–k). Recently, it was reported that the FMRP-R138Q mutant fails to restore activity-dependent bulk endocytosis (ADBE) in Fmr1-KO neurons, perhaps through the loss of BK channel interaction[32]. Our EM measurements also revealed a significant increase in the density of presynaptic vesicles in $Fmr1^{R138Q}$ hippocampal termini (Fig. 2c). This ultrastructural observation could potentially result from an impaired presynaptic glutamate release or a disruption in the exocytosis/recycling process of the synaptic vesicles. This may in turn directly impact the neurotransmitter release and, together with the identified postsynaptic alterations, may mask the effects on the mEPSC frequency in the $Fmr1^{R138Q}$ hippocampus.

The hippocampal plasticity acts on several brain functions that impact a wide range of behavioral responses. We found that the R138Q mutation leads to altered cognitive performances in both male and female $Fmr1^{R138Q}$ mice using the novel object recognition (NOR) test (Fig. 6c, d), likely due to the impaired hippocampal LTP in these mice. Indeed, while it is known that the perirhinal cortex is critical to object recognition memory assessed with the NOR test, it has been proposed that the perirhinal cortex and the hippocampus play complementary roles in the NOR task[33,34]. In particular, at the start of the training session, object information is stored in the perirhinal cortex and when a threshold amount of information is acquired, object information is transferred to the hippocampus and becomes strong object memory. If the threshold is not reached, then the information remains perirhinal cortex dependent as a weak object memory[34]. Thus, it is likely that the impaired LTP observed in the $Fmr1^{R138Q}$ mice participates in the altered cognitive performances observed during the NOR test.

Here, we identified strong social deficits in male $Fmr1^{R138Q}$ mice (Fig. 6f, g). It has been demonstrated that the hippocampal formation is also involved in different components of the social repertoire, including social memory, adaptation to new social contexts, and the maladaptive social behavior observed across several psychiatric disorders[35]. While the hippocampal plasticity defects in the $Fmr1^{R138Q}$ mice may not completely explain the social deficits observed in males, it is likely they contribute, at least partially, to such abnormal behavior. Interestingly, we did not measure any significant impairment in the social behavior of female $Fmr1^{R138Q}$ mice using the three-chamber test (Fig. 6e, f). Sex differences in the function of the hippocampus including the hippocampal neuronal morphology and synaptic plasticity have been observed in rodents[36]. For instance, estrogens may interfere and play a protective role in the $Fmr1^{R138Q}$ social phenotype[37]. Indeed, several preclinical, as well as clinical studies reported a neuroprotective role of estrogens in a number of neurodevelopmental disorders[38–43], including autism[39,40]. However, while estrogens may play a protective role contributing to the normal behavior of the $Fmr1^{R138Q}$ female mice in the three-chamber test, on the basis of the current behavioral data we cannot exclude that $Fmr1^{R138Q}$ female mice still display defects in their social repertoire. Such impairments may be either more subtle in females than in males or revealed by behavioral tasks other than the three-chamber test.

Since most of the knowledge on FXS derives from studies using Fmr1-KO models, it is important to compare the phenotype reported in these models with our data on the $Fmr1^{R138Q}$ mice. Because the R138Q mutation occurs in the FMR1 gene, it has been unequivocally associated with the development of an FXS-like pathology. It is important to stress here that the three unrelated FMRP-R138Q patients show markedly variable phenotypes[11,13–15], indicating that the same mutation leads to different clinical features ranging from mild symptoms to a full, complex classical FXS phenotype. Even if some of the defects measured in $Fmr1^{R138Q}$ animals are different from those measured in Fmr1-KO mice, it is important to notice that they rely on alterations in the same cellular processes, including synaptic elimination and AMPAR-mediated synaptic function, which likely underlie the socio-emotional and cognitive deficits. For instance, an increased spine density and immaturity have been consistently reported in the Fmr1-KO hippocampus[18,19,44,45]. While we also measured an increase in spine density in the $Fmr1^{R138Q}$ hippocampus, we did not notice any modification in the overall maturity of the dendritic protrusions. In addition, while the surface levels of AMPARs are significantly increased in the $Fmr1^{R138Q}$ hippocampus, it is rather decreased in different brain regions of the Fmr1-KO mouse[46–49], including the hippocampus. However, alterations in postsynaptic AMPAR surface expression result in impaired basal synaptic transmission in both models[49] (Fig. 4). Finally, we have shown that the NMDAR-mediated LTP is severely impaired in the $Fmr1^{R138Q}$ hippocampus. While LTP impairments have also been reported in the Fmr1-KO hippocampus[50,51], the molecular mechanisms underlying these defects in $Fmr1^{R138Q}$ mice are likely different from those in the Fmr1-KO background. We also showed that $Fmr1^{R138Q}$ mice display reduced social interaction in the three-chamber test, which is reminiscent of autistic-like features. $Fmr1^{R138Q}$ mice are also unable to discriminate between a familiar and a novel object demonstrating that learning and memory processes are impaired in these animals. Similar cognitive and socio-emotional deficits have been reported in the Fmr1-KO mice in the same behavioral tasks[52,53]. Altogether, our data indicate that different mutations in the FMR1 gene engage distinct molecular mechanisms leading to similar pathological conditions. Therefore, the identification of additional FMRP-R138Q patients will undoubtedly help to shed light on the phenotypic and mechanistic similarities and differences with the classical FXS pathology.

From a therapeutic point of view, it might be of interest to target the AMPARs in the $Fmr1^{R138Q}$ brain. The excess of available AMPARs in the $Fmr1^{R138Q}$ brain may correlate with the intractable seizures observed in the first-reported patient carrying the R138Q mutation[13]. The FDA-approved anti-epileptic AMPAR antagonist Perampanel[54] could be used to reduce the activity of these glutamate receptors and treat the epileptic manifestations in FXS patients carrying the R138Q mutation. Preclinical studies are now required to first investigate the epileptic activity in the $Fmr1^{R138Q}$ mouse line and then determine if Perampanel can correct the altered AMPAR function, as well as the socio-cognitive behaviors in these mice.

Generating preclinical models for brain disorders is an essential step toward the development of efficient therapies to treat human diseases. In this context, the $Fmr1^{R138Q}$ mouse line certainly represents a unique preclinical model to test the efficacy of new molecules and/or the repurposing of existing FDA-approved drugs to correct the altered pathways identified in these mice. This may also lead in the near future to the development of clinical studies to assess the potential benefits of these drugs in FXS patients. Therefore, gaining insights into this complex neurodevelopmental disorder may lead to innovative therapeutic strategies for these particular cases of FXS. Nevertheless, since AMPAR alterations as well as synaptic defects have been linked to other neurodevelopmental disorders, these approaches might be extended to classical FXS patients and more generally to patients presenting ID and ASD in which the AMPAR pathway is altered.

## Methods

**Mouse lines**. $Fmr1^{R138Q}$ mice were successfully generated at the 'Institut Clinique de la Souris' (ICS; Illkirch-Graffenstaden, France) using standard procedures of homologous recombination in murine embryonic C57BL/6N stem (ES) cells (Fig. 1a). $Fmr1^{R138Q}$ mice were backcrossed for more than ten generations into the C57BL/6J genetic background (Janvier, St. Berthevin, France). All animals were handled and treated in accordance with the European Council directives for the Care and Use of Laboratory Animals and following the ARRIVE guidelines. Mice had free access to water and food. Mice were exposed to a 12-h light/dark cycle and the temperature was maintained at $23 \pm 1 \,°C$ with a relative humidity of 45 to 65%.

**Histology and nissl staining**. Brains of PND90 WT and $Fmr1^{R138Q}$ male littermates were carefully dissected and frozen in pre-cooled isopentane in liquid nitrogen ($-60 \,°C$) for 1 min and stored at $-20 \,°C$ until cryostat sectioning. Twenty micrometers section containing the hippocampus were then quickly fixed in 4% formaldehyde in PBS. Hippocampal sections were then Nissl stained to reveal the gross architecture of the hippocampus. Fixed sections were first immersed in a cresyl violet solution ($5 \, g \, L^{-1}$ cresyl violet, 0.3% acetic acid) for 2–3 min, then quickly washed in distilled water, and 70% ethanol/0.05% acetic acid. After dehydrating in 100% ethanol, sections were mounted with Entellan and imaged with a 4x lens on a Leica DMD optical microscope.

**Golgi-cox staining**. WT and $Fmr1^{R138Q}$ male littermates at PND90 were deeply anesthetized by an intraperitoneal injection of $50 \, mg \, kg^{-1}$ sodium pentobarbital. Mice were then transcardially perfused with a 0.9% NaCl (w/v) solution to remove blood from the vessels. Brains were stained with the FD Rapid GolgiStain Kit (FD NeuroTechnologies) following the manufacturer's instructions. Briefly, tissues were quickly rinsed in ddH$_2$O and immersed in the impregnation solution for 14 days at RT in the dark. Brains were then transferred to the tissue-protecting solution and kept in the dark at RT for a further 72 h. Hundred-micrometers-coronal section mounted in the tissue-protecting solution on coated slides (2% gelatin, 1% KCr (SO$_4$)$_2$) were then air dried at room temperature (RT) for 2 h in the dark. After two 4-min washes in ddH$_2$O, sections were stained in an ammonium and sodium thiosulfate-containing solution for 10 min. Sections were then dehydrated successively in ethanol with increasing percentage (50 to 100%) and finally, in xylene for 4 min and mounted in Entellan. Images were acquired in the following 48 h using an Axiovert200M videomicroscope (Zeiss) with a 100x oil immersion lens. Z-series of randomly selected basal and apical secondary dendrites from CA1 hippocampal neurons were processed with the Extended Depth of Field plugin of ImageJ. Dendritic spine length, width, and density were measured in ImageJ and data imported in GraphPad Prism software for statistical analysis.

**Electron microscopy**. WT and $Fmr1^{R138Q}$ male littermates at PND90 were deeply anesthetized by an intraperitoneal injection of $50 \, mg \, kg^{-1}$ sodium pentobarbital. Mice were transcardially perfused with a 0.9% NaCl solution and then with 2.5% glutaraldehyde and 2% paraformaldehyde in 0.15 M sodium cacodylate buffer (pH 7.4). Brains were post fixed for an additional 24–48 h at $4 \,°C$. Hippocampi were then manually dissected from 100 µm thick vibratome sections. After washing, they were further fixed in 2% osmium tetroxide, rinsed, stained with 1% uranyl acetate in water for 45 min, dehydrated, and embedded in epoxy resin (Electron Microscopy Science, Hatfield, PA, USA). Ultrathin 70–90-nm sections were stained with uranyl acetate and lead citrate, then imaged under a Philips CM10 transmission electron microscope (TEM; FEI, Eindhoven, Netherlands). Images were acquired at a final magnification of 25–34000× using a Morada CCD camera (Olympus, Munster, Germany). Excitatory synapses in the apical dendrite layer of the hippocampal CA1 region were selected for analyses based on the presence of a cluster of presynaptic vesicles, a defined synaptic cleft, and an electron-dense postsynaptic density (PSD). The density of excitatory synapses and presynaptic vesicles, as well as the length and thickness of the PSD, were computed. The average thickness of the PSD was calculated as described in[55,56]. The estimation of the density of excitatory synapses per µm$^3$ of the hippocampus was computed using a size-frequency stereological method[56,57].

**Mouse brain lysate preparation**. Brain lysates from WT and $Fmr1^{R138Q}$ littermates from the indicated developmental stages (PND3–90) were prepared. Briefly, freshly dissected brains were transferred in five volumes (w/v) of ice-cold sucrose buffer (10 mM Tris-HCl pH 7.4, 0.32 M sucrose) supplemented with a protease inhibitor cocktail (Sigma, 1/100) and homogenized at $4 \,°C$ using a Teflon–glass potter. Nuclear fraction and cell debris were pelleted by centrifugation at $1000 \, g$ for 10 min. The post nuclear S1 fraction (supernatant) was collected and protein concentration measured using the BCA protein assay (Biorad).

**Total mRNA analysis**. Total RNA was extracted from PND90 WT and $Fmr1^{R138Q}$ brains with the Trizol reagent (Sigma) according to the manufacturer's recommendations. RNA was then purified using the RNeasy Mini Kit (Qiagen). RT was performed on 1 µg of RNA with the Superscript IV synthesis kit (Invitrogen). Quantitative PCR (qPCR) was performed on a Light Cycler 480 (Roche) with MasterMix SYBRGreen (Roche) using specific oligonucleotides (Supplementary Table 1) and following the manufacturer's instructions.

**Primary neuronal cultures**. The protocol to prepare primary neuronal cultures from mouse embryos was approved by the National Animal Care and Ethics Committee (Project reference APAFIS#18648-2019011111154666). Hippocampal neurons were prepared from WT and $Fmr1^{R138Q}$ embryonic (E15.5) C57BL/6 mice. Neurons were plated in Neurobasal medium (Invitrogen, France) supplemented with 2% B27 (Invitrogen), 0.5 mM glutamine, and penicillin/streptomycin (Ozyme) on 60-mm dishes or 16-mm glass coverslips (VWR) pre-coated with Poly-L-Lysine ($0.5 \, mg \, mL^{-1}$; Sigma). Neurons (600,000 cells per 60-mm dish or 80,000 cells per 16-mm coverslip) were then used at 14–15 DIV.

**Sindbis virus production and neuronal transduction**. Attenuated Sindbis viral particles (SINrep(nsP2S726)) were prepared by generating cRNAs from the pSin-Rep5 plasmid (Invitrogen) containing the sequence coding for eGFP and from the defective helper (pDH-BB) plasmid[58,59] using the Mmessage Mmachine SP6 solution (Ambion). Both cRNAs were then electroporated into BHK21 cells. Pseudovirions present in the culture medium were harvested 48 h after electroporation and ultracentrifuged on a SW41Ti. Aliquots of concentrated Sindbis particles were titrated and stored at $-80 \,°C$ until use. Neurons were transduced at a multiplicity of infection (MOI) of 0.1 and incubated at $37 \,°C$ under 5% CO$_2$ for 20 h until use.

**Biotinylation assays[60]**. Live Tetrodotoxin (TTX, 0.5 µM)-treated control or treated $Fmr1^{R138Q}$ hippocampal neurons (14–15 DIV) were surface biotinylated for 10 min at $4 \,°C$ on ice using the membrane impermeant Sulfo-NHS-SS-Biotin (Pierce, $0.3 \, mg \, mL^{-1}$ in PBS). After three washes in ice-cold PBS, neurons were incubated with NH$_4$Cl (50 mM in PBS) for 5 min at $4 \,°C$ to quench the remaining unbound biotin. After three more washes in PBS, cells were lysed in extraction buffer (Tris-HCl 10 mM pH 7.5, NaCl 150 mM, EDTA 10 mM, Triton X-100 1%, SDS 0.1%, mammalian protease inhibitor cocktail 1%) for 1 h at $4 \,°C$. After sonication and centrifugation ($16,000 \, g$ for 15 min at $4 \,°C$), supernatants containing an equal amount of protein were incubated with 50 µl streptavidin beads for 3 h to overnight at $4 \,°C$ to immunoprecipitate the surface-biotinylated proteins. After four washes in extraction buffer, proteins were eluted from the streptavidin beads by boiling in reducing sample buffer containing 5% β-mercaptoethanol and then resolved by SDS-PAGE and immunoblotting using the indicated antibodies. Standard GAPDH controls were included to ensure that there was no intracellular biotinylation. Bands were quantified using ImageJ and normalized as indicated in the figure legends.

**BS3-crosslinking assays[56]**. Hippocampi from WT and $Fmr1^{R138Q}$ male littermates were included in agar. Two hundred fifty-micrometer-thick hippocampal slices were prepared in ice-cold oxygenated (5% CO$_2$, 95% O$_2$) sucrose solution (2.5 mM KCl, 1.25 mM NaH$_2$PO$_4$, 10 mM MgSO$_4$, 0.5 mM CaCl$_2$, 26 mM NaHCO$_3$, 11 mM glucose, 234 mM sucrose). Free-floating hippocampal slices were first pretreated with TTX in ECS solution (140 mM NaCl, 1.3 mM CaCl$_2$, 5 mM KCl, 25 mM HEPES, 33 mM glucose, Tris buffered to pH 7.4, TTX 1 µM) for 10 min at $37 \,°C$ and then incubated in ECS or cLTP[21] solutions. LTP was chemically induced with 200 µM glycine, 20 µM bicuculline, and 1 µM strychnine in pre-warmed ECS for 3 min. After a 20-min washout in ECS, slices were incubated in the cell membrane-impermeable BS3 crosslinker (PierceNet) that was prepared as a 52 mM stock solution in 5 mM sodium citrate buffer pH 5 and added onto hippocampal slices at a final concentration of 2 mM, for 30 min at $4 \,°C$ with gentle agitation. Glycine (100 mM) was then added for 10 min at $4 \,°C$ to quench the remaining unbound BS3. Slices were homogenized in lysis buffer (50 mM Tris-HCl pH 7.5, 150 mM NaCl, 1 mM EDTA, 1% SDS, mammalian protease inhibitor cocktail 1%) for 1 h at RT. After sonication and centrifugation at $16,000 \, g$ for 15 min, protein concentration was measured using the BCA protein assay (Bio-Rad). Proteins (20 µg) were resolved on 7% acrylamide SDS-PAGE gels and blotted for GluA1 and GluA2. Standard β3-tubulin controls were included to ensure that intracellular proteins were not BS3 crosslinked. Bands were quantified using ImageJ software (NIH). A surface/intracellular ratio was performed to analyze the levels of surface GluA1 and GluA2 expression.

**Immunoblotting**. Protein extracts were resolved by SDS-PAGE, transferred onto nitrocellulose membrane (BioTraceNT, PALL), immunoblotted with the following primary antibodies: mouse anti-FMRP[18] 1 µg/mL (DSHB, Clone 2F5-1); Rabbit anti-GluA1 C-terminal 1/1000 (Merk-Millipore #AB1504); Rabbit anti-GluA2 C-terminal 1/2000 (Synaptic System #182103); Rabbit anti-Homer1 1/1000 (Synaptic Systems #160003); Mouse anti-PSD95 1/10000 (NeuroMab #75-028 clone K28/43); Goat anti-GluN1 1/500 (Santa-Cruz #sc-1467); Mouse anti-Synapsin1a/b 1/500 (Santa-Cruz #sc-376623); Rabbit anti-GAD65/67 1/500 (Merk-Millipore #ABN904); Rabbit anti-CaMKII 1/500 (Santa-Cruz #sc-9035); Rabbit anti-Arc 1/1000 (Synaptic Systems #156003); Mouse anti-Gephyrin 1/1000 (Synaptic System #147111); Rabbit anti-vGluT1 1/5000 (Synaptic System #135303). Standard loading controls were included using a rabbit anti-GAPDH antibody 1/25000 (Sigma #G9545) or a rabbit anti-β3 Tubulin 1/25000 (Synaptic Systems #302302) as indicated. Proteins were revealed using the appropriate HRP-conjugated secondary antibodies (GE healthcare #NA931V and #NA934V). Proteins were then identified using Immobilon Western (Millipore) chemiluminescent solution and images

acquired on a Fusion FX7 system (Vilber Lourmat). Full-size blots for cropped gels are shown in the Source data file.

**Immunolocalization of surface-expressed GluA1 AMPAR subunits.** Live TTX-treated WT and $Fmr1^{R138Q}$ hippocampal neurons (14–15 DIV) were incubated with a mouse monoclonal anti-N-terminal GluA1 antibody (1:100; Merk-Millipore #MAB2263) for 10 min at RT. After three washes, the neurons were quickly fixed in PBS containing 3.7% formaldehyde and 5% sucrose for 5 min at RT. Cells were then thoroughly rinsed and incubated with a non-permeabilizing blocking solution (0.2% BSA, 5% HS in PBS) for 30 min. Neurons were then stained for 1 h at RT with the appropriate secondary antibodies (Thermofisher/Invitrogen #A-21202 and #A-21203; 1:200) conjugated to Alexa488 or 594 as indicated in PBS containing 0.2% BSA. Neurons were then further fixed for 10 min in PBS containing 3.7% formaldehyde and 5% sucrose to stabilize the surface-associated secondary antibodies. After a 30-min blocking/permeabilizing step in PBS containing 0.2% BSA, 0.2% Triton X-100, and 5% HS, neurons were incubated with guinea pig anti-MAP2 (1/1000; Synaptic systems #188004) antibodies overnight at 4 °C. Cells were then washed three times in PBS and incubated with the indicated secondary Alexa-conjugated antibodies for 1 h at RT, mounted with Mowiol (Sigma), and stored at −20 °C until confocal examination. The overall surface GluA1 intensity which represents the fluorescence associated with the GluA1 antibody labeling was then measured, as well as the density of surface GluA1 clusters, which rather correlates with the distribution of the surface-expressed AMPARs at synapses along dendrites since GluA1 is generally concentrated to dendritic spines.

For STED imaging, live TTX-treated neurons were incubated with either a mouse monoclonal anti-N-terminal GluA1 (1/40, Merk-Millipore #MAB2263) or GluA2 (1/30, Merk-Millipore #MAB397) antibodies and treated or not for cLTP as above. To visualize the surface-expressed subunits, neurons were labeled with the appropriate secondary antibodies conjugated to the StarRED dye (Abberior GmbH STRED-1001; 1/100) prior to the permeabilization step and staining for Homer1 with a rabbit anti-Homer1 (1/500; Synaptic System). Coverslips were mounted in Abberior Mount Solid Antifade (Abberior GmbH, Göttingen). STED images were acquired using a Leica SP8 STED 3X (Leica Microsystems, Nanterre), at 400 Hz through a 100x/1.4 NA Oil objective using the Leica LAS X software. Z-stack of confocal images of Homer1 immunolabelled with Alexa594-labeled secondary antibodies were obtained by a laser excitation at 561 nm and combined with a STED image acquisition of surface GluA1 or GluA2 immunostained with a StarRed Fluorophore excited at 633 nm and depleted at 775 nm (20–30% of power). Z-stack of 2D STED images had a $20 \times 20 \times 200$-nm voxel size and all images were deconvolved using Huygens Professional (v18.10, Scientific Volume Imaging, The Netherlands) and the CMLE algorithm with respectively SNR:20 and SNR:14 for the confocal and STED images with 40 iterations.

**Analysis of synaptic AMPAR nanodomains.** Deconvolved Z-stack images of confocal Homer1-Alexa594/STED GluA1/2-StarRed were analyzed using a home-made macro program in FIJI software[48] to quantify the levels of surface-expressed GluA1 and GluA2 on dendritic spines. Deconvolved Z-stacks were first converted to 2D images by a sum projection. The Homer1 image was then segmented (Median filtering, intensity thresholding, Watershed) to delineate mature dendritic spines with a diameter criterion (>250 nm). Corresponding Spine Regions of Interest (SROI) obtained was first used to measure the intensity of the surface GluA1/2 staining below this SROI on GluA1/2 images. Then the number of nanodomains per spine was counted by an individual SROI screening on GluA1/2 images: each SROI was duplicated to give a small surface GluA1/2 image per spine and segmented. The number of surface-associated GluA1 and GluA2 nanodomains detected per spine in secondary dendrites was incremented according to a size criterion, assuming a maximum size of 90 nm. Data were then imported in GraphPad Prism software for statistical analysis.

**Imaging and analysis of spine morphology.** WT and $Fmr1^{R138Q}$ hippocampal neurons were transduced at 14 DIV with Sindbis virus (MOI of 0.1) expressing free eGFP for 20 h before fixation in PBS containing 3.7% formaldehyde and 5% sucrose for 20 min at RT. Cells were then mounted in Mowiol as above. Confocal images were acquired through a 63x oil objective (numerical aperture NA 1.4) on a laser scanning confocal microscope LSM780 (Carl Zeiss, Marly-le-Roy, France). The 488-nm laser power and detection gain (510–550 nm range) were adjusted to avoid signal saturation. Z-stacks of three to five images of randomly selected GFP-expressing secondary dendrites were compressed into two dimensions using a maximum projection with the ImageJ software. Maximal Intensity Projection images were imported into NeuronStudio[61] for the automatic detection of the dendritic spines. The length of individual spines was automatically measured and data imported in GraphPad Prism software for statistical analysis.

**mEPSC patch-clamp recordings.** Mice were anesthetized (Ketamine 150 mg kg⁻¹ / Xylazine 10 mg kg⁻¹) and transcardially perfused with aCSF for slice preparation. Two hundred fifty-micrometer-thick acute transverse hippocampal slices were prepared. In ice-cold dissecting solution (234 mM sucrose, 2.5 mM KCl, 0.5 mM CaCl₂, 10 mM MgCl₂, 26 mM NaHCO₃, 1.25 mM NaH₂PO₄, and 11 mM D-glucose) oxygenated with 95% O₂ and 5% CO₂ at pH 7.4. Slices were first incubated

for 1 h at 37 °C in aCSF (119 mM NaCl, 2.5 mM KCl, 1.25 mM NaH₂PO₄, 26 mM NaHCO₃, 1.3 mM MgSO₄, 2.5 mM CaCl₂, and 11 mM D-glucose) oxygenated with 95% O₂ and 5% CO₂ at pH 7.4. Slices were used after recovering for another 30 min at RT. Visualized whole-cell voltage-clamp recording techniques were used to measure AMPAR miniature excitatory postsynaptic currents (mEPSCs), using an upright microscope (Olympus France). Whole-cell recordings were performed using a Multiclamp 700B (Molecular Devices, UK) amplifier, under the control of pClamp10 software (Molecular Devices, UK). Slices were kept at 32–34 °C in a recording chamber perfused with 2.5 ml min⁻¹ aCSF (119 mM NaCl, 2.5 mM KCl, 1.25 mM NaH₂PO₄, 26 mM NaHCO₃, 2 mM MgSO₄, 4 mM CaCl₂, and 11 mM D-glucose) oxygenated with 95% O₂ and 5% CO₂. Picrotoxin (50 μM; Sigma-Aldrich, France) to block GABAergic transmission and TTX (0.5 μM; Abcam, France) to block sodium channels were added to the aCSF perfusion solution. Recording pipettes (5–6 MΩ) for voltage-clamp experiments were filled with a solution containing 117.5 mM Cs-gluconate, 15.5 mM CsCl, 10 mM TEACl, 8 mM NaCl, 10 HEPES, 0.25 mM EGTA, 4 mM MgATP, and 0.3 NaGTP (pH 7.3; Osmolarity 290–300 mOsm). Whole-cell patch-clamp configuration was established at a Voltage holding (Vh) of −65 mV, and cells were left to stabilize for 2–3 min before recordings. Holding current and series resistance were continuously monitored throughout the recordings and, if either of these parameters varied by more than 20%, the experiment was discarded. At least 100 events were obtained from each cell. Experiments and analysis were done blind to the genotype. Recordings were analyzed using the Clampfit software by applying a threshold search to detect spontaneous events. The threshold for AMPAR mEPSCs detection was set at −9 pA to exclude electric noise contamination.

**Slice preparation and LTP electrophysiological recordings.** WT and $Fmr1^{R138Q}$ male littermate (PND35–42) brains were incubated for 4 min in ice-cold sucrose cutting solution (19.99 mM KCl, 260 mM NaHCO₃, 11.5 mM NaH₂PO₄, 10 mM glucose, 220 mM sucrose, 0.2 mM CaCl₂, 6 mM MgCl₂) gassed with carbogen (95% O₂ + 5% CO₂; pH 7.4 at 37 °C; 290–310 mosmol L⁻¹). Sagittal hippocampal slices (350 μm) were then incubated in standard artificial cerebrospinal fluid (ACSF; 119 mM NaCl, 2.5 mM KCl, 1.3 mM MgCl₂, 2.5 mM CaCl₂, 10 mM glucose, 1 mM NaH₂PO₄, 26 mM NaHCO₃). Slices were allowed to recover at 35 °C for 45 min, continuously oxygenated with carbogen, and then for 45 min at RT. Recordings were performed in ACSF continuously oxygenated with carbogen at 33–35 °C in a submerged recording chamber perfused at a low rate (1.8 ml min⁻¹). Glass microelectrodes (tip diameter 5–10 μm; resistance: 0.2–0.3 Ω) were filled with ACSF. Field recordings were performed using MultiClamp 700B amplifier (Molecular Devices, Foster City, CA, USA) and Clampfit software. Schaffer collaterals in the CA1 region were stimulated using a bipolar electrode and recorded in the *stratum radiatum* of the CA1. A baseline of 10 min was recorded in the current-clamp mode with a single stimulation at 0.1 Hz every 10 s. LTP was induced by high-frequency stimulation ($3 \times 100$ Hz for 1 s, 30 s inter burst), and recorded at 0.1 Hz stimulation rate for 50 min. The analysis was performed using the Clampfit software (Molecular Device). Along with the 0.1 Hz stimulations, fiber volley (FV) and synaptic response slopes were calculated and normalized to the baseline levels. Experiments were done blind to the genotype. To assess for synaptic changes, the EPSP/FV slope ratio was calculated and averaged for 1-min time periods. Quantification and statistical comparisons were computed by comparing the EPSP/FV ratios obtained at 40–50 min after induction to the baseline level.

**Behavioral tasks.** All experiments were performed between 9:00 a.m. and 4:30 p.m. Both male and female WT and $Fmr1^{R138Q}$ mice were tested. The experiments were approved by the Italian Ministry of Health (Rome, Italy; Authorization number: 87-2019-PR) and performed in agreement with the ARRIVE guidelines, with the guidelines released by the Italian Ministry of Health (D.L. 26/14) and the European Community Directive 2010/63/EU. All behavioral experiments were performed and scored blind to the genotype. Data are expressed as mean ± s.e.m.

*Locomotor activity.* At PND40–45, during the habituation session of the novel object recognition test, locomotor activity was calculated using a grid that divides the arena into equally sized squares and that is projected over the recordings. The number of line crossings made by the animal was quantified to assess its motor activity.

*The Isolation-induced ultrasonic vocalizations (USVs) test.* The test was performed as previously described[19]. Briefly, each pup (PND7) was individually removed from the nest and placed into a black Plexiglas arena, located inside a sound-attenuating and temperature-controlled chamber. USVs from the pups were detected for 3 min by an ultrasound microphone (Avisoft Bioacoustics, Germany) sensitive to frequencies between 10 and 250 kHz placed at 10 cm above the arena. Pup axillary temperature was measured before and after the test by a digital thermometer. The emission of USVs was analyzed using Avisoft Recorder software (Version 5.1).

*Novel object recognition test.* The novel object recognition test was performed at PND40–45. The test consisted of three phases: habituation, training, and test. In the habituation phase, the animals were allowed to explore an empty arena (a Plexiglas arena measuring $40 \times 40 \times 40$ cm³) for 5 min. Twenty-four hours later, on

the training trial, each mouse was individually placed into the arena containing two identical objects (A1 and A2), equidistant from each other, and allowed to explore the objects for 10 min. After 1 h, during the test phase, one copy of the familiar object (A3) and a new object (B) was placed in the same location as during the training trial. The time spent exploring each object was recorded for 5 min. The discrimination index was calculated as the difference in time exploring the novel and the familiar objects, expressed as the percentage ratio of the total time spent exploring both objects[62].

*Three-chamber test.* This test was performed at PND40–45. The apparatus was a rectangular three-chamber box, with two lateral chambers (20.5 (length) × 41 (width) × 22.5 (height) cm) connected to a central chamber (20 (length) × 41 (width) × 22.5 (height) cm). Each lateral chamber contained a small Plexiglas cylindrical cage. Each experimental mouse was individually allowed to explore the central compartment for 5 min with both doors closed. Next, each experimental mouse was individually allowed to explore the apparatus for 10 min and then confined in the central compartment. An unfamiliar stimulus animal was confined in a cage located in one chamber of the apparatus, while the cage in the other chamber was left empty. Both doors to the side chambers were then opened, allowing the experimental animal to explore the apparatus for 10 min. The percentage of time spent in the social approach (sniffing the stimulus animal) and the discrimination index were scored using the Observer XT, version 12.0 software (Noldus Information Technology, The Netherlands). The discrimination index was calculated as the difference in time exploring the stimulus and the empty cage, expressed as the percentage ratio of the total time spent exploring both the stimulus and empty cage.

**Data manipulation and statistical analysis**. Statistical analyses were calculated using GraphPad Prism (v7.0; GraphPad Software, Inc) or Sigma Plot (v13.0; Systat Software, Inc, USA) software. All data are expressed as mean ± s.e.m. Unpaired *t* test (Figs. 2c, 3a, 4, 5c, d, k and Supplementary Figs. 1b and 3), Ratio *t* test (Figs. 3c, d, 5b, h and Supplementary Fig. 4) or non-parametric Mann–Whitney test (Figs. 2a, b, 3b, 5e and supplementary Fig. 2) were used to compare medians of two data sets. Statistical significance for multiple comparison data sets was computed two-way ANOVA with Sidak's post test (Fig. 1b). Behavioral data (Fig. 6 and Supplementary Fig. 6) were analyzed by two-way ANOVA including genotype and sex as factors with a Newman–Keuls post hoc test for individual group comparisons. Normality for all groups was verified using the Shapiro–Wilk test. For electrophysiological data, distributions were analyzed by a Kolmogorov–Smirnov test (Fig. 4). *$p < 0.05$ was considered significant.

**Reporting summary**. Further information on research design is available in the Nature Research Reporting Summary linked to this article.

## Data availability
The datasets generated and analyzed during the current study are available from the corresponding author on reasonable request. The B6N *Fmr1*$^{R138Q}$ line is available to the scientific community via the Mouse Clinical Institute (ICS, Illkirsh, France; http://www.ics-mci.fr/en/), or the INFRAFRONTIER consortium (the European Research Infrastructure for phenotyping and archiving of model mammalian genomes (https://www.infrafrontier.eu/). Source data are provided with this paper.

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

## Acknowledgements

We gratefully acknowledge the 'Fondation pour la Recherche Médicale' (Equipe labellisée #DEQ20140329490 to BB and 'Fin de thèse' #FDT202012010480 to MP), the 'Agence Nationale de la Recherche' (ANR-20-CE16-0006-01 to SM, ANR-20-CE16-0006-02 to YH, and ANR-20-CE16-0016-01 to BB), the 'Jérôme Lejeune' (SM, VT), 'Fondation pour la Recherche sur le Cerveau' (AO2018, BB), the IDEX UCA^Jedi interdisciplinary master program (CG), and 'Bettencourt-Schueller' (SM) foundations for financial support. We also thank the French government for the 'Investments for the Future' LabEx 'SIGNA-LIFE' (ANR-11-LABX-0028-01), LabEx 'ICST' (ANR-11-LABX-0015-01), and the IDEX UCA^Jedi ANR-15-IDEX-01, as well as the CG06 (AAP santé), GIS IBiSA (AO 2014), and 'Région Provence-Alpes-Côte d'Azur' for the 'Microscopy and Imaging Côte d'Azur' (MICA) platform funding. MPri is a fellow from the international Ph.D. 'SIGNALIFE' program.

## Author contributions

M. Prieto performed Golgi assays and all the spine density analysis in hippocampal cultures and slices. A.F., M. Pronot, and M. Prieto achieved all biotinylation and crosslinking assays. A.F., M. Prieto, M. Pronot, and G.P. performed all biochemical experiments on total lysates. A.F., M. Pronot, and M. Prieto performed all the immunocytochemistry. M. Prieto and G.P. monitored the growth and fertility of the *Fmr1^R138Q* mouse line. A.F., G.P., M. Pronot, and M. Prieto prepared neuronal cultures. G.P. purified primary antibodies and with A.F., M. Prieto, M.C., and M. Pronot, performed some hippocampal slice preparations. S.A. imaged the STED experiments. F.B. provided computational tools to analyze imaging data. P.P. performed and analyzed mEPSCs experiments in hippocampal slices. M.C. and E.D. performed some electrophysiological recordings. M. Prieto prepared brains for EM experiments. N.L. and M.F. performed and analyzed EM. V.B. and S.S. performed and analyzed behavioral experiments. V.T. provided guidance and analysis of behavioral data. U.F. performed the electrophysiological synaptic plasticity experiments and Y.H. analyzed the data and supervised the work. S.C. performed RNA experiments. B.B. provided specific mRNA probes and RNA work supervision. A.K. and C.G. performed some initial experiments. M. Prieto, A.F., and S.M. contributed to hypothesis development, experimental design, and data interpretation. S.M. provided the overall supervision, the funding, and wrote the original draft. All authors commented on the manuscript. S.M. edited the manuscript.

## Competing interests

The authors declare no competing interests.
