## [Peer Review File · Nature Communications]

Reviewers' Comments:

Reviewer #1:

Remarks to the Author:

In this manuscript, the authors generate a mouse model for a form of Fragile X syndrome (FXS) caused by a mutation (R138Q) in FMRP. This mouse has behavioral phenotypes that are consistent with FXS, and has cell biological phenotypes (increased spine density and surface expression of AMPARs) and physiological phenotypes (deficits in LTP) that could explain the behavior. The data presented are convincing and novel, and provide an exciting new insight into the molecular mechanisms that underlie this form of X-linked intellectual disability. There are a number of points that require clarification with additional discussion, and I also suggest some additional imaging data should be presented, as outlined below.

The KI animals show only a change in total GluA1; there is no change in total expression of GluA2. In many pathological scenarios (eg brain ischemia), this would result in the synaptic expression of Ca²⁺ permeable AMPARs. However, in most of the experiments performed in this manuscript, surface GluA2 is affected to a similar extent as GluA1. Can the authors discuss this apparent discrepancy?

The EPSC amplitude is higher in the mutant animals, while the frequency is unaffected. This seems surprising since an increase in spine density might be expected to manifest as an increase in frequency. Does this suggest some of the "extra" spines do not contain functional synapses (presumably not, since the EM shows an increase in synapse density)? This should be discussed further.

The authors separately measure both surface intensity and cluster density. What do these two parameters define differentially? This should be explained in the manuscript.

It is unclear what the STED experiment in fig 4a contributes to the paper, beyond what is shown in earlier figures. Do the authors propose that the super-res clusters are nano-domains within single synapses? If so, what is the relevance of the increase in this parameter specifically? The authors should be able to use these images to define the mean synaptic GluA1 signal intensity, which strikes me as being a more relevant parameter. Also, is GluA2 affected between WT and KI in this assay?

Given the discrepancy between the biochemistry/ cell imaging and the electrophysiology in Fig.5, can the authors perform STED imaging on cultures treated for cLTP and determine whether this treatment causes a loss of synaptic GluA1?

In the discussion, the authors state, "The direct consequence of this plasticity defect is an abnormal socio-cognitive behaviour". As the data stand, this statement goes beyond the data presented, since a direct link has not been demonstrated. It is definitely an important question: how are the cell biological and synaptic plasticity effects in the hippocampus related to the behavioral phenotypes? Object recognition memory involves mainly LTD in the perirhinal cortex, rather than hippocampal LTP and presumably the social interaction deficits (Fig 6e/f) also do not involve hippocampal LTP. I appreciate that the hippocampal preparations (slices and cultures) are used as a model for analysing synaptic function in general, which I believe is absolutely appropriate, but additional discussion is needed to clarify this point.

Furthermore, the behavioral experiments (Fig 6) were carried out on both male and female animals, whereas all other experiments appear to be restricted to males. Since the social interaction phenotype is restricted to male animals, if there is a direct link between the plasticity defects and the socio-cognitive behaviour, it might be expected that the synaptic and/or AMPAR differences are differentially expressed in males compared to females. This should also be discussed further in the manuscript.

In some figure legends, the sample size (n number) is missing. This should be shown for all experiments.

Fig 6A seems unnecessary and could be removed.

Reviewer #2:

Remarks to the Author:

Silencing and loss-of-expression of the Fragile X Mental Retardation Protein (FMRP) leads to Fragile X syndrome (FXS), the most frequent form of inherited intellectual disability. This work addresses functional deficiencies triggered by one Fmr1 gene mutation found in patients, FMRP-R138Q. To this aim, the authors have reproduced the mutation in mice and characterized atypical synaptic plasticity and autistic-like behaviors of this knock-in mouse model (Fmr1R138Q). They found an increased spine density, with postsynaptic ultrastructural defects and increased AMPA receptor surface expression in the hippocampus of these mice. This is associated with long-term potentiation (LTP) and socio-cognitive deficits. All together, these results unveil how R138Q mutation affects the postsynaptic function of FMRP and its impact on mice behavior. The work is overall of high interest, performed with thoroughness, and well written. Please find below specific comments that should be address before acceptance for publication.

1- The authors compared the total levels of a subset of FMRP target mRNAs in PND90 WT and Fmr1R138Q male littermate brains by RT-qPCR (Fig. 1c) and found no significant differences in the total mRNA levels of the FMRP targets tested.

How was the "subset" of gene chosen? What about gene coding for proteins which could control AMPA-R cell surface expression (even those which mRNA are not FMRP binders could nevertheless be indirectly misexpressed in presence of FMR1-R138Q)? For example, the mRNA levels for GRIP1, PICK1 or TARPs should be checked, because this could explain the increased synaptic surface expression of AMPA receptor (Fig. 4a)

2- How could you explain that the increase in spine density (Figure 2) is not associated with an increase of mEPSCs frequency (Fig. 4d)? Could you test or at least discuss possible compensatory mechanisms? Could it be that the PSD thickness, reduced in Fmr1-R138Q hippocampal neurons prevents adequate PSD functionality (could you site publications in favor of this hypothesis)? Alternatively, a reduced presynaptic release of Glutamate could be involved? On this topic, it is mentioned in the discussion "In contrast, the present data did not reveal any obvious physiological impairment linked to the presynaptic function in the Fmr1R138Q mouse model." Which precise experiment do you refer to?

3- Given the high increase in spine density (Figure 2), I would have expected an increase in PSD95 expression as well (Fig. 3a), could you comment, please? As an alternative to the experiment suggested in 1, the protein level of GRIP1 and PICK1 could be assessed rather than mRNA levels.

4- Figure 3b (and 5e), what is the interpretation of an increased "cluster density" of AMPA receptors? What is the spatial resolution? What is the unit? What do you call "cluster"?

5- Figure 4a, in STED experiments, the mean synaptic surface GluA1 cluster density is significantly increased in KI mice compared to WT mice. You may here refer to nanoclusters? What is the functional consequences expected from a cluster density increase? It is stated "These data indicate that the missense R138Q mutation leads to a significant increase in available postsynaptic AMPARs in the Fmr1R138Q hippocampus". Do larger cluster means more AMPA receptors? Could you site specific references measuring the number of AMPA receptors depending on cluster size? On the other hand, Homer1 positive clusters seem larger in KI hippocampal neurons as well. Could you please quantify? If this last assumption is confirmed then the number of AMPA-R per Homer1-

positive area would not be changed? Again how could we interpret these findings in terms of receptor activity and synaptic transmission efficiency?

Homer staining is rather diffuse, which might be because "Homer" is a confocal image. If so, this should be written on the picture, please.

6- Fig. 5 shows an impaired LTP in the KI. While the protocol to induce LTP unexpectedly decreases the number of cell surface receptors in the KI, electrical recordings show no change in synaptic transmission efficiency, no LTP but no LTD either. Please suggest hypothesis to explain this discrepancy (while AMPA receptor cell surface expression is reduced, no LTD is recorded).

7- mGlu5-dependent excessive LTD is a well described feature of FMR1 KO. This work should be completed with experiments assessing whether LTD can be induced by mGlu5 stimulation in the KI mice hippocampus? Is it excessive as well?

Thank you for taking these comments into account.

Sincerely,

Julie Perroy

Responses to reviewers

Reviewer #1

We are grateful to this referee for his/her enthusiasm towards our work and for the insightful comments and constructive suggestions. S/he states: *'In this manuscript, the authors generate a mouse model for a form of Fragile X syndrome (FXS) caused by a mutation (R138Q) in FMRP. This mouse has behavioral phenotypes that are consistent with FXS, and has cell biological phenotypes (increased spine density and surface expression of AMPARs) and physiological phenotypes (deficits in LTP) that could explain the behavior. The data presented are convincing and novel, and provide an exciting new insight into the molecular mechanisms that underlie this form of X-linked intellectual disability. There are a number of points that require clarification with additional discussion, and I also suggest some additional imaging data should be presented, as outlined below.'*

As suggested, we carried out additional imaging experiments and discussed all the points raised which greatly improved the quality of the manuscript. The reviewer's comments are in blue.

1. The KI animals show only a change in total GluA1; there is no change in total expression of GluA2. In many pathological scenarios (eg brain ischemia), this would result in the synaptic expression of Ca²⁺ permeable AMPARs. However, in most of the experiments performed in this manuscript, surface GluA2 is affected to a similar extent as GluA1. Can the authors discuss this apparent discrepancy?

We thank the reviewer for raising this point and we agree that an increase in total GluA1 but not GluA2 levels may suggest changes in AMPAR subunit composition. In the initial version of the manuscript, we showed that total GluA2 levels were slightly higher in the *Fmr1^{R138Q}* brain but not statistically significant compared to WT controls. These data were obtained comparing 3 sets of WT and *Fmr1^{R138Q}* male littermates. We therefore decided to increase the N number to 6 littermates to strengthened our analysis (**Revised figure 3a**). Despite this increase in the N number, we still did not observe any significant differences in the total GluA2 levels between WT and *Fmr1^{R138Q}* brains, in line with our original report.

We showed that the overall surface expression of both GluA1 and GluA2 AMPAR subunits is significantly increased in the *Fmr1^{R138Q}* hippocampus, suggesting that the R138Q mutation enhances AMPAR expression at the plasma membrane. As suggested by the reviewer, we performed additional super resolution microscopy (STED) to measure and compare the levels of post-synaptic surface-expressed AMPAR subunits in WT and *Fmr1^{R138Q}* neurons (**Revised figure 4a-d**). Using this approach, we measured important differences between the two genotypes. In particular, we observed that the postsynaptic density of both surface GluA1 and GluA2-containing nanodomains is increased in *Fmr1^{R138Q}* hippocampal neurons in line with the increase in the amplitude observed in the mEPSCs recordings. Interestingly, we also detected an increase in the post-synaptic mean fluorescence levels of surface GluA1 and a decrease in GluA2 in *Fmr1^{R138Q}* neurons. This result probably indicates that, contrary to GluA1, the surface increase in GluA2 subunits rather occurs extrasynaptically.

Altogether, these data suggest an alteration in the distribution, the targeting and a differential trafficking of AMPAR in the *Fmr1^{R138Q}* hippocampus. We have now implemented the manuscript with these new imaging data and discussed these points in the revised manuscript.

2. The EPSC amplitude is higher in the mutant animals, while the frequency is unaffected. This seems

surprising since an increase in spine density might be expected to manifest as an increase in frequency. Does this suggest some of the “extra” spines do not contain functional synapses (presumably not, since the EM shows an increase in synapse density)? This should be discussed further.

We agree with the reviewer that the mEPSC frequency usually correlates with spine/synapse density, but also with the presynaptic release probability. We found that *Fmr1*^{R138Q} neurons show an increase in synapse density in the *Fmr1*^{R138Q} hippocampus, which could result in an increased mEPSC frequency. However, we cannot exclude that the R138Q mutation may also affect the presynaptic compartment. Therefore, we analysed the ultrastructure of the presynaptic compartments using our EM experiments (**Revised Fig. 2c**). We measured a significant increase in the density of synaptic vesicles in presynaptic *Fmr1*^{R138Q} terminals compared to their WT controls highlighting potential alterations in the presynaptic function of *Fmr1*^{R138Q} neurons.

Furthermore, the authors of a recent preprint from BioRxiv (Bonnycastle et al., <https://www.biorxiv.org/content/10.1101/2020.09.10.291062v1.full>) report that the exogenous expression of FMRP-R138Q mutant in *Fmr1*-KO neurons fails to restore the defective activity-dependent bulk endocytosis (ADBE). ADBE is the dominant mechanism for synaptic vesicle (SV) retrieval during elevated neuronal activity and thus, impairment of this process in *Fmr1*^{R138Q} neurons may explain the increased density of synaptic vesicles measured in our EM experiments.

Therefore, the absence of an increased mEPSC frequency in the *Fmr1*^{R138Q} hippocampus might just result from a combination of altered pathways both in the pre- and post-synaptic compartments. We have now discussed this point in the revised manuscript and additional work will now be needed to clarify the precise impact of the R138Q mutation on the presynaptic function *in vivo*.

3. The authors separately measure both surface intensity and cluster density. What do these two parameters define differentially? This should be explained in the manuscript.

The surface intensity represents the fluorescence associated with the overall surface AMPAR antibody labelling. In our model, this parameter is significantly increased, thus revealing an overall increase in AMPAR surface expression in *Fmr1*^{R138Q} hippocampal neurons.

The cluster density is a complementary measurement and rather refers to the distribution of the surface-expressed AMPARs along dendrites. Since GluA1 is generally localized to dendritic spines, we wanted to confirm whether the increased spine density measured in *Fmr1*^{R138Q} neurons in both Golgi and EM experiments (**Revised figure 2**) correlates with an increase in the density of surface GluA1 clusters (**Revised figure 3b**).

This is now indicated in the method section of the revised manuscript.

4-1. It is unclear what the STED experiment in fig 4a contributes to the paper, beyond what is shown in earlier figures. Do the authors propose that the super-res clusters are nano-domains within single synapses? If so, what is the relevance of the increase in this parameter specifically? The authors should be able to use these images to define the mean synaptic GluA1 signal intensity, which strikes me as being a more relevant parameter. Also, is GluA2 affected between WT and KI in this assay?

We thank the reviewer for raising this important point. We do agree that the term cluster we used in super-resolution experiments is not appropriate. We have now exchanged this word for ‘nanodomain’ in the revised manuscript, which is more adequate and in line with the current literature in the field of AMPARs. Indeed, post-synaptic AMPARs are organized into 80-90 nm nanodomains facing the presynaptic glutamate release sites to allow an efficient synaptic transmission (MacGillavry et al.

2013 PMID 23719161, Nair et al. 2013 PMID 23926273). Given the low affinity of AMPARs for glutamate, the synaptic nanoscale AMPAR organization is essential to synaptic strength. Indeed, glutamate release over clustered receptors induced an increase in the amplitude of the post-synaptic response (MacGillavry et al. 2013 PMID 23719161). In addition, super-resolution microscopy data revealed a transsynaptic nanocolumn organization involving a complex of presynaptic proteins essential to SV dynamics facing PSD95 clusters at the post-synapse (Tang et al. 2016 PMID 27462810, Reddy-Alla et al. 2017 PMID 28867551, Chamma et al. 2016 PMID 26979420).

Thus, as suggested by the reviewer, we performed additional STED imaging experiments on WT and *Fmr1^{R138Q}* hippocampal cultures to precisely measure the surface expression and the number of nanodomains containing the available AMPARs at the post-synaptic membrane. We have now included the analyses in the revised manuscript and compared the synaptic surface expression of both GluA1 and GluA2 (**Revised figure 4a-c**) in WT and *Fmr1^{R138Q}* hippocampal neurons. These data revealed a significant increase in post-synaptic surface GluA1 fluorescence and a decrease in surface-associated GluA2 fluorescence indicating that the increased surface level of GluA2 measured in biochemical experiments is rather due to its extrasynaptic increase highlighting defects in the trafficking of AMPARs.

Interestingly, we also quantified the density of surface-expressed GluA1 and GluA2-containing nanodomains in the *Fmr1^{R138Q}* hippocampus (**Revised figure 4a,b,d**). Consistent with the literature (Nair et al. 2013 PMID 23926273), we measured a density of ~2-2.1 nanodomains per spine for both surface-associated GluA1 and GluA2 in WT hippocampal neurons while there was a significant increase in the mean number of both surface-associated GluA1 and GluA2 nanodomains in *Fmr1^{R138Q}* synapses indicating that the R138Q FXS mutation not only impact the surface expression of GluA1 and GluA2 but also differentially perturbs the synaptic trafficking of AMPARs in *Fmr1^{R138Q}* neurons. However, since there is an increase in the density of surface GluA2-containing nanodomains concomitant with a decrease in their surface GluA2 fluorescence, we cannot exclude the possibility that the overall amount of available GluA2 per individual synapse remains similar to the WT. We have now included and discussed these new findings in the revised version of the manuscript.

4-2. Given the discrepancy between the biochemistry/ cell imaging and the electrophysiology in Fig. 5, can the authors perform STED imaging on cultures treated for cLTP and determine whether this treatment causes a loss of synaptic GluA1?

We have now performed the suggested STED experiments to measure and compare the mean synaptic GluA1 intensity and the density of GluA1-containing nanodomains in WT and *Fmr1^{R138Q}* hippocampal neurons upon the induction of cLTP (**Revised figure 5f,g**). As expected, we observed a significant increase in both the number of synaptic surface GluA1-containing nanodomains and their mean synaptic GluA1 fluorescence intensity upon cLTP in WT synapses, indicating that LTP was properly induced. Conversely, the number of synaptic GluA1-containing nanodomains remained unchanged in *Fmr1^{R138Q}* synapses upon cLTP, whereas its mean synaptic surface GluA1 fluorescence intensity was decreased (**Revised figure 5f,g**). Therefore, we can now suggest that the combination of the decreased level of available surface GluA1 subunits without affecting the number of post-synaptic nanodomains in cLTP along with a lower PSD thickness and the altered density of synaptic vesicles and spines (**Revised figure 2**) in the *Fmr1^{R138Q}* brain participate in a dramatic impairment of the hippocampal LTP (**Revised figure 5i-k**). We have now discussed these new data extensively in the revised version of the manuscript.

5a. In the discussion, the authors state, “The direct consequence of this plasticity defect is an

abnormal socio-cognitive behaviour". As the data stand, this statement goes beyond the data presented, since a direct link has not been demonstrated. It is definitely an important question: how are the cell biological and synaptic plasticity effects in the hippocampus related to the behavioral phenotypes? Object recognition memory involves mainly LTD in the perirhinal cortex, rather than hippocampal LTP and presumably the social interaction deficits (Fig 6e/f) also do not involve hippocampal LTP. I appreciate that the hippocampal preparations (slices and cultures) are used as a model for analysing synaptic function in general, which I believe is absolutely appropriate, but additional discussion is needed to clarify this point.

We agree with the reviewer that this sentence may sound a bit too speculative since a link between the hippocampal plasticity defects and the altered socio-cognitive behaviour displayed by the *Fmr1^{RI38Q}* mice can only be hypothesized in the present work. It is known that the hippocampal plasticity acts on different brain functions that impact on a wide range of behavioral responses. This is particularly true for the memory function. The novel object recognition (NOR) test is a useful tool to evaluate the discrimination capacity of rodents, and thus their ability to recognize a previously known object from a novel one. Although several studies have indicated that the perirhinal cortex plays a pivotal role in encoding, consolidation and retrieval of object recognition memory (reviewed for instance in Winters et al., 2008 PMID: 18499253), a model in which both the perirhinal cortex and the hippocampus act in tandem in object recognition memory during NOR has also been proposed (Cohen et al., 2015, PMID: 25169255). According to this model, at the start of the training session, object information is stored in the perirhinal cortex. After a threshold amount of object information is acquired, object information is transferred to the hippocampus. If this threshold is not reached, then the information will remain perirhinal cortex-dependent as a weak object memory. Conversely, if the threshold is reached, then the information will become hippocampal-dependent as a strong object memory (Cohen et al., 2015, PMID: 25169255). Thus, it is likely that the impaired LTP measured in *Fmr1^{RI38Q}* mice participates in the altered cognitive performances identified during the NOR test. We have now discussed this point in the revised version of the manuscript.

Regarding the social interaction deficits in the *Fmr1^{RI38Q}* mice, it is widely accepted that the hippocampus has a role in different components of the social repertoire. For instance, it supports social memory, allows adaptation to new social contexts, and it is involved in the maladaptive social behaviour observed across several psychiatric disorders (Montagrin et al., 2018 PMID: 28843041). However, we agree with the reviewer that the current data do not prove a direct link between plasticity defects in the hippocampus and abnormal sociability. On these bases, we have now mitigated our conclusions about a direct link between hippocampal plasticity and the socio-cognitive deficits found in the *Fmr1^{RI38Q}* mice in the revised version of the manuscript.

5b. Furthermore, the behavioral experiments (Fig 6) were carried out on both male and female animals, whereas all other experiments appear to be restricted to males. Since the social interaction phenotype is restricted to male animals, if there is a direct link between the plasticity defects and the socio-cognitive behaviour, it might be expected that the synaptic and/or AMPAR differences are differentially expressed in males compared to females. This should also be discussed further in the manuscript.

We have performed additional BS3-crosslinking assays on hippocampal slices obtained from WT and *Fmr1^{RI38Q}* female mice (**Reviewer figure 1**). These data show similar defects in the hippocampal response to cLTP in *Fmr1^{RI38Q}* female than in male (**figure 5c,d; Supplementary figure 3d,e**) with an absence of surface increase in AMPAR in the *Fmr1^{RI38Q}* background.

Importantly, the sex-related differences found in the three-chamber test (**figure 6e,f**) are not necessarily linked to a difference in the synaptic expression of AMPAR in *Fmr1*^{R138Q} female mice. Indeed, sex differences in hippocampus functioning have been observed in several mammalian species and studies examining sex differences in hippocampal neuronal morphology and synaptic plasticity in rodents exist (reviewed for instance in Kos et al. 2016 PMID: 27870401). However, different explanations can account for the social “recover” of female *Fmr1*^{R138Q} mice. For instance, estrogens could interfere with the social phenotype exerting a protective role in females (Pinares-Garcia et al. 2018 PMID: 30104506). In line with this possibility, studies from both preclinical and clinical research suggest a neuroprotective role of estrogens in a number of neurodevelopmental disorders (Crider et al. 2017 PMID: 27789681) including autism (Gockley et al. 2015 PMID: 25973162; Lai et al. 2015 PMID: 25973161). Moreover, accumulating evidence demonstrates that sex chromosome genes can directly influence brain function in normal and pathological conditions (Dewing et al. PMID: 16488877; Dewing et al. PMID: 14559357), thus concurring to the normal behaviour of *Fmr1*^{R138Q} female mice in the three-chamber test.

In addition, estrogens have been repeatedly shown to influence a wide array of social behaviours at multiple levels including the detection and integration of socially relevant olfactory information to more complex social behaviors like social preference and learning and memory for social stimuli (Ervin et al. 2015 PMID: 26122289). For these reasons, it cannot be excluded that a certain impairment in the social repertoire may still be present in *Fmr1*^{R138Q} female mice, in accordance with the socio-communicative deficits displayed at infancy (**figure 6a,b**), but such impairments could be either more subtle in females or emerge in other behavioural tasks than the three-chamber test. On these bases, we have now discussed the sex-related differences identified in *Fmr1*^{R138Q} mice in the revised version of the manuscript.

[REDACTED]

6. In some figure legends, the sample size (*n* number) is missing. This should be shown for all experiments.

Fig 6A seems unnecessary and could be removed.

- We thank the reviewer for pointing out the missing sample size in some figure legends. We have now added all the N numbers in the revised manuscript.
- We believe the USV diagram in Fig. 6A makes the behavioural set up more comprehensive so we prefer to keep it in the figure.

Reviewer #2

We are grateful to this referee for his/her enthusiasm towards our work and thank him/her for the insightful comments and constructive suggestions. S/he states: *'Silencing and loss-of-expression of the Fragile X Mental Retardation Protein (FMRP) leads to Fragile X syndrome (FXS), the most frequent form of inherited intellectual disability. This work addresses functional deficiencies triggered by one Fmr1 gene mutation found in patients, FMRP-R138Q. To this aim, the authors have reproduced the mutation in mice and characterized atypical synaptic plasticity and autistic-like behaviors of this knock-in mouse model (Fmr1R138Q). They found an increased spine density, with postsynaptic ultrastructural defects and increased AMPA receptor surface expression in the hippocampus of these mice. This is associated with long-term potentiation (LTP) and socio-cognitive deficits. All together, these results unveil how R138Q mutation affects the postsynaptic function of FMRP and its impact on mice behavior. The work is overall of high interest, performed with thoroughness, and well written. Please find below specific comments that should be address before acceptance for publication.'*

We carried out additional experiments to answer the points made and discussed the new data in the revised manuscript. The reviewer's comments are in blue.

1a. The authors compared the total levels of a subset of FMRP target mRNAs in PND90 WT and Fmr1R138Q male littermate brains by RT-qPCR (Fig. 1c) and found no significant differences in the total mRNA levels of the FMRP targets tested. How was the "subset" of gene chosen?

Given that FMRP can bind more than 800 mRNAs in the brain (Maurin et al. 2018 PMID: 29668986), we have chosen to focus on some of the best-characterized mRNA targets of FMRP that are important to the synaptic function.

1b. What about gene coding for proteins which could control AMPA-R cell surface expression (even those which mRNA are not FMRP binders could nevertheless be indirectly misexpressed in presence of FMR1-R138Q)? For example, the mRNA levels for GRIP1, PICK1 or TARPs should be checked, because this could explain the increased synaptic surface expression of AMPA receptor (Fig. 4a)

Along with the second part of the referee Point 3.** Given the high increase in spine density (Figure 2), I would have expected an increase in PSD95 expression as well (Fig. 3a), could you comment, please? **As an alternative to the experiment suggested in 1, the protein level of GRIP1 and PICK1 could be assessed rather than mRNA levels.

We agree with the referee that it is of interest to compare the levels of proteins impacting the surface expression of AMPAR to clarify the defects observed in the *Fmr1^{R138Q}* mice. We have thus decided to perform additional experiments as suggested in the referee's point 3, to measure and compare the total levels of GRIP1, PICK1 and Stargazin proteins in WT and *Fmr1^{R138Q}* male littermates, instead of their mRNA expression levels, since they are not direct targets of FMRP. As shown in the *new*

figure 3a, we did not measure any significant differences in PICK1, GRIP1 nor Stargazin protein levels between the two genotypes indicating that the altered surface expression and trafficking of AMPAR in *Fmr1^{R138Q}* neurons is not due to an altered expression of these accessory proteins. These new data are now included (**Revised figure 3a**) and discussed in the revised version of the manuscript.

2. How could you explain that the increase in spine density (Figure 2) is not associated with an increase of mEPSCs frequency (Fig. 4d)? Could you test or at least discuss possible compensatory mechanisms? Could it be that the PSD thickness, reduced in Fmr1-R138Q hippocampal neurons prevents adequate PSD functionality (could you cite publications in favor of this hypothesis?)? Alternatively, a reduced presynaptic release of Glutamate could be involved? On this topic, it is mentioned in the discussion “In contrast, the present data did not reveal any obvious physiological impairment linked to the presynaptic function in the Fmr1R138Q mouse model.” Which precise experiment do you refer to?

This point is similar to the second point of the Referee 1 and we agree with both reviewers that the mEPSC frequency usually correlates with spine/synapse density, which is increased in *Fmr1^{R138Q}* neurons (**figure 2**). However, mEPSCs are also linked to the presynaptic function. We have now included a comparative ultrastructural analysis of the presynaptic compartments of the WT and *Fmr1^{R138Q}* hippocampus in the revised version of the manuscript. These data revealed a significant increase in the density of synaptic vesicles in presynaptic *Fmr1^{R138Q}* terminals (**Revised figure 2c**) thus indicating that the R138Q mutation also affects the presynaptic compartment which, in turn, could directly impact the glutamate release.

As indicated in the answer to the third point raised by the Referee 1, a recent preprint reports that the exogenous expression of the FMRP-R138Q mutant fails to restore activity-dependent bulk endocytosis (ADBE) in *Fmr1*-KO neurons (Bonnycastle et al <https://www.biorxiv.org/content/10.1101/2020.09.10.291062v1.full>). Since ADBE, the main mechanism for synaptic vesicle (SV) retrieval, is still impaired in KO neurons overexpressing FMRP-R138Q FXS mutant, it may thus explain the increased density of SV measured in hippocampal synapses using EM (**Revised figure 2c**).

We also agree with the reviewer that it is likely that the reduced PSD thickness measured in *Fmr1^{R138Q}* hippocampal synapses participates in the impaired synaptic function. Indeed, a reduced PSD thickness has been reported in several Knock-out mouse models targeting proteins relevant to the synaptic function where these ultrastructural alterations correlate with synaptic transmission, plasticity and behavioural defects (Hung et al. 2008 PMID: 18272690; Peça et al. 2011 PMID: 21423165; Zhu et al. 2015 PMID: 25751059).

Therefore, we do believe that the mEPSCs data (**Figure 4i-k**) which represent the overall response of *Fmr1^{R138Q}* hippocampal neurons, result from an increased spine density along with alterations in both the pre- (*SV density*) and post-synaptic (*PSD thickness, AMPAR surface expression and nanoscale distribution*) compartments. We have now implemented the manuscript with these new data and discussed these points in the revised version of the manuscript.

3. Given the high increase in spine density (Figure 2), I would have expected an increase in PSD95 expression as well (Fig. 3a), could you comment, please? As an alternative to the experiment suggested in 1, the protein level of GRIP1 and PICK1 could be assessed rather than mRNA levels.

We agree with the referee that an increase in PSD95 protein levels could be expected given the increased spine density in the *Fmr1^{R138Q}* brain. However, we showed in EM experiments (**Figure 2c**)

that the PSD thickness is significantly reduced in *Fmr1^{R138Q}* hippocampal synapses, which likely underlies alterations in the PSD protein composition. Since we did not observe any differences in the total level of PSD95 protein (**Figure 3a**), we can speculate that the reduction of the PSD thickness might underlie a reduced PSD95 protein level per synapse. This reduction is probably compensated by the increased spine density measured in *Fmr1^{R138Q}* mice (**Figure 2**).

For the second part of the question (As an alternative to the experiment suggested in 1, the protein level of GRIP1 and PICK1 could be assessed rather than mRNA levels.), please see our answer in point 1b.

4. Figure 3b (and 5e), what is the interpretation of an increased “cluster density” of AMPA receptors? What is the spatial resolution? What is the unit? What do you call “cluster”?

We thank the referee for these questions that are in line with the third point of Referee 1. The cluster density in the original figures 3b and 5e refers to the distribution of the surface-expressed AMPARs along dendrites. Since GluA1 is generally concentrated in dendritic spines, we wanted to confirm whether the increased spine density measured in *Fmr1^{R138Q}* neurons in both Golgi and EM experiments correlates with an increase in the density of surface GluA1 clusters. Given that both figures 3b and 5a represent confocal images, the xy resolution for the surface AMPAR clusters detected in these experiments is around 300 nm. There is no specific unit for the measurements made in figures 3b and 5b since it corresponds to a normalisation of the WT and *Fmr1^{R138Q}* values to their basal conditions. Prior to their normalization, the cluster density was determined as the number of surface-labelled AMPAR clusters per unit of dendritic length. This is now better explained in the revised manuscript.

*5-1. Figure 4a, in STED experiments, the men synaptic surface GluA1 cluster density is significantly increased in KI mice compared to WT mice. You may here refer to nanoclusters? What is the functional consequences expected from a cluster density increase? It is stated “These data indicate that the missense R138Q mutation leads to a significant increase in available postsynaptic AMPARs in the *Fmr1^{R138Q}* hippocampus”. Do larger cluster means more AMPA receptors? Could you site specific references measuring the number of AMPA receptors depending on cluster size? On the other hand, Homer1 positive clusters seem larger in KI hippocampal neurons as well. Could you please quantify? If this last assumption is confirmed then the number of AMPA-R per Homer1-positive area would not be changed? Again how could we interpret these findings in terms of receptor activity and synaptic transmission efficiency?*

The referee is right and his/her point is in line with the point 4 of Reviewer 1. We do agree that the term ‘cluster’ is not appropriate for our STED data. We have now exchanged this term for ‘nanodomain’ in the revised manuscript. This is particularly important since post-synaptic AMPARs are organized into 80-90 nm nanodomains facing the presynaptic glutamate release sites to allow an efficient synaptic transmission (MacGillavry et al. 2013 PMID 23719161, Nair et al. 2013 PMID 23926273). As stated above (*Ref 1 point 4-1*), the synaptic nanoscale AMPAR organization is essential given their low affinity for glutamate, to allow the receptors to sit in front of the glutamate releasing sites. For instance, it has been shown that the glutamate release over clustered AMPARs leads to an increase in the amplitude of the synaptic responses (MacGillavry et al. 2013 PMID 23719161) indicating that the clustering and nanoscale organisation of AMPARs at the post-synapse is central to synaptic transmission and plasticity. Using STED experiments, we measured an increase in the overall AMPAR nanodomains in *Fmr1^{R138Q}* neurons which likely concentrates the available AMPARs in front of presynaptic releasing sites, thus enhancing synaptic transmission, as observed

in mEPSC recordings in *Fmr1^{R138Q}* neurons (**Revised figure 4i-k**).

As requested by the referee, we have now quantified and compared the mean surface area for Homer1 clusters in WT and *Fmr1^{R138Q}* neurons (**Referee fig. 2**). The mean surface of post-synaptic Homer1 sites is significantly increased in *Fmr1^{R138Q}* neurons (Normalized to WT: WT: 1 ± 0.035 ; *Fmr1^{R138Q}*: 1.24 ± 0.047). The referee is right that if we normalize the number of surface-expressed AMPAR clusters with the size of the Homer1 clusters, the differences between WT and *Fmr1^{R138Q}* neurons may be masked. However, we quantified the number of nanodomains containing surface-expressed AMPARs in Homer-labelled synapses and we did not normalize to the Homer1 area. We think that the increase in absolute number of surface-expressed AMPAR-containing nanodomains per synapse better reflects the synaptic changes in the nanoscale organization of AMPARs.

We have now reorganized the manuscript including these new sets of super-resolution imaging experiments and thoroughly rearranged the result and discussion sections in the revised version of the manuscript.

5-2. Homer staining is rather diffuse, which might be because “Homer” is a confocal image. If so, this should be written on the picture, please.

The referee is right that Homer1 staining is a confocal image, as written in the figure legend. We have now added this information on STED images as well, to avoid any misunderstanding.

6. Fig. 5 shows an impaired LTP in the KI. While the protocol to induce LTP unexpectedly decreases the number of cell surface receptors in the KI, electrical recordings show no change in synaptic transmission efficiency, no LTP but no LTD either. Please suggest hypothesis to explain this discrepancy (while AMPA receptor cell surface expression is reduced, no LTD is recorded).

We thank the reviewer for raising this point, which relates to point 4-2 raised by Referee 1. As above-mentioned, we performed the suggested surface GluA1 STED imaging experiments upon cLTP induction in WT and *Fmr1^{R138Q}* hippocampal neurons. We found that, as expected, the number of synaptic GluA1-containing AMPAR clusters as well as the mean synaptic GluA1 intensity are both significantly increased in WT synapses upon cLTP. Interestingly, the number of synaptic surface GluA1-containing nanodomains remained unchanged in *Fmr1^{R138Q}* synapses upon cLTP, whereas the mean synaptic GluA1 intensity was decreased (**Revised figure 5c,d**). These data thus indicate that the trafficking and synaptic regulation of AMPARs is altered in *Fmr1^{R138Q}* neurons during LTP. Therefore, we have now discussed these STED data in relation to the dramatic impairment of hippocampal LTP measured in slice electrophysiological experiments (**Revised figure 5i-k**). We now suggest that the combination of the altered post-synaptic AMPAR trafficking and nanoscale synaptic distribution in cLTP along with the lower PSD thickness and the increased density of presynaptic vesicles and dendritic spines (**Revised figure 2**) in the *Fmr1^{R138Q}* brain participate in the plasticity defects measured in hippocampal slices. These new data are now extensively discussed in the revised version of the manuscript.

7. mGlu5-dependent excessive LTD is a well described feature of FMR1 KO. This work should be completed with experiments assessing whether LTD can be induced by mGlu5 stimulation in the KI mice hippocampus? Is it excessive as well?

We agree with the referee that, given the excessive mGlu5R-dependent LTD in *Fmr1*-KO mice (Huber et al. 2002 PMID: 12032354; Bear et al. 2004 PMID: 15219735), it could be of interest to evaluate the impact of the R138Q mutation in LTD. To be consistent with the literature in *Fmr1*-KO mice, we thus performed recordings of excitatory postsynaptic field potentials on acute hippocampal slices obtained from PND26-31 WT and *Fmr1*^{R138Q} littermates in mGlu5R-stimulated conditions (**Reviewer figure 3**). LTD was chemically induced with 100 μ M (S)-3,5-dihydroxyphenyl-glycine ((S)-3,5-DHPG), a selective agonist of group I mGlu1/5 receptors. We found that the mGlu1/5R-dependent LTD is similar in both genotypes indicating that LTD is not impaired in the *Fmr1*^{R138Q} hippocampus at this developmental stage, unlike in *Fmr1*-KO mice. To date, we do not have any biochemical, nor classical imaging or super-resolution data related to this process in WT and *Fmr1*^{R138Q} hippocampal neurons and slices, and we do think that many additional experiments should be performed to clearly state that there are no defects in the LTD process in the *Fmr1*^{R138Q} genotype. Therefore, we kindly ask the referee to keep these data unpublished as it may lead to a complete separate set of exciting data when thoroughly controlled. In addition, we do believe that this LTD experiment will not change the overall conclusions of the present study.

[REDACTED]

[REDACTED]

Concluding statement: We do believe that the additional sets of data provided in the revised version of the manuscript as well as the points of discussion suggested by both expert reviewers greatly improved the quality of the manuscript. We therefore think this revised work will generate intense interest worldwide in the fields of Neurosciences and Cell Biology by providing additional resources to neuroscientists to better assess the pathophysiological relevance of the *FMRI* R138Q missense mutation in FXS.

Reviewers' Comments:

Reviewer #1:

Remarks to the Author:

In this revised version, the authors have improved the manuscript with additional discussion/clarifications and some extra data. However, there are a couple of points that still need to be addressed prior to publication.

I still feel that aspects of the AMPAR subunit data don't quite add up. On P.9 of the revised manuscript, the authors state, "the upregulation of surface GluA2 measured in biochemical experiments is rather due to its extrasynaptic increase". If this is correct, it would strongly suggest that Ca-permeable AMPARs must be expressed at R138Q synapses. This point must be discussed. I don't think there is any mention of Ca²⁺-permeable (CP-)AMPARs in the manuscript. Moreover, the data show that there is a greater number of synaptic clusters containing GluA2 in the R138Q neurons compared to WT, which again supports the argument that on average, a R138Q cluster must have less GluA2 compared to WT. In summary, there are still clear discrepancies in the subunit-specificity data that need to be discussed in the context of CP-AMPARs.

Example images for the STED experiments in fig.5f,g should be included.

Reviewer #2:

Remarks to the Author:

My concerns have been addressed in the revisions. Thank you for this work.

Responses to reviewers

The reviewer's comments are in blue.

Reviewer #1

The referee has one last point of discussion regarding the revised version of our manuscript. S/He states: *'In this revised version, the authors have improved the manuscript with additional discussion/clarifications and some extra data. However, there are a couple of points that still need to be addressed prior to publication.*

I still feel that aspects of the AMPAR subunit data don't quite add up. On P.9 of the revised manuscript, the authors state, "the upregulation of surface GluA2 measured in biochemical experiments is rather due to its extrasynaptic increase". If this is correct, it would strongly suggest that Ca-permeable AMPARs must be expressed at R138Q synapses. This point must be discussed. I don't think there is any mention of Ca²⁺-permeable (CP-)AMPARs in the manuscript. Moreover, the data show that there is a greater number of synaptic clusters containing GluA2 in the R138Q neurons compared to WT, which again supports the argument that on average, a R138Q cluster must have less GluA2 compared to WT.

In summary, there are still clear discrepancies in the subunit-specificity data that need to be discussed in the context of CP-AMPARs.

Example images for the STED experiments in fig.5f,g should be included.'

We thank the reviewer for his/her insightful comments. We agree that the data presented in the revised manuscript could lead to the hypothesis that the intrinsic composition in AMPAR subunits at the postsynapse is altered in *Fmr1^{R138Q}* neurons. As suggested by this reviewer, we have now discussed this possibility and that the synaptic content in Calcium permeable/impermeable AMPAR subunits might be different between the genotypes (*page 16, discussion, red section*).

We have also included GluA1 STED images for the cLTP condition in figure 5.

Reviewer #2

The reviewer states: *'My concerns have been addressed in the revisions. Thank you for this work.'*

We are grateful to this referee for his/her strong support toward our work.

Reviewers' Comments:

Reviewer #1:

Remarks to the Author:

The authors have satisfactorily addressed my concerns, and I have no further criticisms.

Response to Reviewer 1

The reviewer's comment is in blue.

Reviewer #1

The reviewer states: *The authors have satisfactorily addressed my concerns, and I have no further criticisms.*

We thank the referee for his/her strong support toward our work.